# Follicular T cells are clonally and transcriptionally distinct in B cell-driven mouse autoimmune disease

Elliot H. Akama-Garren [1,2], Theo van den Broek [1], Lea Simoni[1], Carlos Castrillon[1], Cees E. van der Poel[1] & Michael C. Carroll [1✉]

Pathogenic autoantibodies contribute to tissue damage and clinical decline in autoimmune disease. Follicular T cells are central regulators of germinal centers, although their contribution to autoantibody-mediated disease remains unclear. Here we perform single cell RNA and T cell receptor (TCR) sequencing of follicular T cells in a mouse model of autoantibody-mediated disease, allowing for analyses of paired transcriptomes and unbiased TCRαβ repertoires at single cell resolution. A minority of clonotypes are preferentially shared amongst autoimmune follicular T cells and clonotypic expansion is associated with differential gene signatures in autoimmune disease. Antigen prediction using algorithmic and machine learning approaches indicates convergence towards shared specificities between non-autoimmune and autoimmune follicular T cells. However, differential autoimmune transcriptional signatures are preserved even amongst follicular T cells with shared predicted specificities. These results demonstrate that follicular T cells are phenotypically distinct in B cell-driven autoimmune disease, providing potential therapeutic targets to modulate autoantibody development.

[1] Program in Cellular and Molecular Medicine, Boston Children's Hospital, Harvard Medical School, Boston, MA 02115, USA. [2] Harvard-MIT Health Sciences and Technology, Harvard Medical School, Boston, MA 02115, USA. ✉email: michael.carroll@childrens.harvard.edu

Autoantibodies are characteristic of many autoimmune diseases, including Sjogren's syndrome, Hashimoto thyroiditis, granulomatosis with polyangiitis, myasthenia gravis, and systemic lupus erythematosus (SLE). These pathogenic autoantibodies bind self-antigen, forming immune complexes that damage host organs or directly damage host tissues, leading to organ failure and clinical decline[1]. Longitudinal studies of patient sera have demonstrated that the specificity of this autoimmune response changes over time to react to a changing set of self-antigens, termed epitope spreading[2]. Although this phenomenon is well documented clinically, the molecular origins of autoantibody responses remain unknown.

Our group has developed a mixed bone marrow chimera model of autoantibody-mediated disease in mice[3]. Reconstitution of irradiated wild type mice with bone marrow containing a single autoreactive B cell clone is capable of initiating multiorgan autoimmune disease. These mice develop autoantibodies not only against the initial self-antigen, ribonucleoprotein, but also an array of unrelated self-antigens such as A proliferation-inducing ligand (APRIL), bactericidal permeability increasing protein (BPI), and glomerular basement membrane (GBM), leading to glomerular deposition of autoantibodies. These autoantibodies derive not from the original autoreactive B cell clone, but from wild type cells which enter germinal center reactions that eventually become independent of the original autoreactive B cell. This entry is T cell-dependent, as administration of anti-CD40L antibodies abrogates recruitment of wild type B cells to germinal centers. The nature of this T cell help, in particular the mechanisms by which follicular T cells contribute to loss of B cell tolerance and clonal evolution towards self-antigens, remains uncharacterized.

Follicular helper T ($T_{FH}$) and follicular regulatory T ($T_{FR}$) cells are critical regulators of the germinal center reaction, promoting the somatic hypermutation and class switch recombination necessary for high affinity antibody responses and B cell memory. In the light zone of germinal center reactions, $T_{FH}$ cells provide CD40 and IL-21 stimulation to B cells to selectively expand high affinity B cell clones, while B cells present antigen and ICOSL stimulation to $T_{FH}$ cells to stabilize the $T_{FH}$ cell fate, reflected by expression of Bcl-6, CXCR5, and PD-1[4–6]. $T_{FR}$ cells derive from natural regulatory T (Treg) cells that have gained expression of CXCR5 and lost expression of CCR7, leading to entry and participation in the germinal center reaction[7,8]. We hypothesized that follicular T cells regulate autoantibody development, and in the presence of a single autoreactive B cell clone adopt a dysfunctional phenotype that permits autoantibody development.

To elucidate possible mechanisms of follicular T cell dysfunction that permit loss of peripheral B cell tolerance, we isolate and perform paired single-cell RNA sequencing (scRNA-seq) and T cell receptor (TCR) sequencing of follicular T cells in a bone marrow chimera model of autoantibody-mediated disease. Analysis of gene expression paired with clonotypic identity indicates distinct patterns of repertoire expansion and accompanying transcriptional changes in autoimmune mice. Furthermore, prediction of antigen binding using VDJ database annotation shows that this differential expression is preserved even amongst clonotypes with the same predicted specificities. These results demonstrate that in the presence of an autoreactive B cell clone follicular T cells adopt a transcriptionally distinct phenotype that is reflected in the TCR repertoire.

## Results

### Autoimmune follicular T cells are transcriptionally distinct.

B-cell driven autoimmune disease is established by reconstituting irradiated wild type mice with congenic wild type (WT) bone marrow mixed with bone marrow from 564Igi mice[3], which have heavy and light chain knock in of an autoreactive B cell receptor against ribonuclear complexes[9]. To better characterize follicular T cells in this process, we performed paired scRNA-seq and scTCR-seq on CD4$^+$CXCR5$^+$PD-1$^+$ cells isolated from mixed 564Igi chimeras ($n = 5$) ten weeks after reconstitution (Fig. 1a). For non-autoimmune controls, we used WT bone marrow chimeras ($n = 5$) that were immunized with NP-OVA six weeks after reconstitution to generate germinal centers against a foreign antigen. After concatenation of data across individual chimeras and quality control, we retained 13,432 cells with 1,653 median genes per cell from 564Igi (autoimmune) chimeras and 15,271 cells with 1,624 median genes per cell from non-autoimmune chimeras (Supplementary Table 1).

We confirmed the follicular T cell identity of sequenced cells by visualizing expression of canonical markers of $T_{FH}$ and $T_{FR}$ cells, observing low expression of Ccr7 and widespread expression of Cd4, Cxcr5, Pdcd1, and Icos (Supplementary Fig. 1). Unsupervised clustering of the combined samples revealed six conserved clusters and we visualized expression of the most significantly differentially expressed genes (DEGs) per cluster (Fig. 1b, Supplementary Data 1). DEGs assigned follicular T cell clusters to the known Foxp3$^+$ $T_{FR}$, Tnfsf8$^+$ activated $T_{FH}$, and Sostdc1$^+$ $T_{FH}$ subtypes, as well as previously undescribed follicular T cells subtypes of Sox4$^+$Sell$^+$ central memory $T_{FH}$, Ccl5$^+$Gzmk$^+$ effector $T_{FH}$, and Ifit1$^+$ interferon stimulated (ISG) $T_{FH}$ (Fig. 1c). Each subset was present in every individual chimera and their relatedness was visualized by hierarchical cluster tree (Supplementary Fig. 1). Pseudotime analysis illustrated potential developmental trajectories between follicular T cell subsets, revealing that central memory, effector, and ISG subsets represent distinct states in pseudotime and in developmental trajectory represented by a diffusion map (Fig. 1d). Gene-pseudotime correlation analysis revealed similar patterns of differentiation in follicular T cells from autoimmune and non-autoimmune chimeras (Supplementary Fig. 1), with decreasing expression of Izuom1r and increasing expression of Nkg7 and Klf2 in pseudotime (Fig. 1e). The distribution of follicular T cells to individual clusters was similar between autoimmune and non-autoimmune chimeras, with the exception of a decrease in central memory $T_{FH}$ cells in autoimmune chimeras (Fig. 1f). We validated this finding by flow cytometry, observing a decrease in the frequency of CD44$^+$CD62L$^+$ cells and compensatory increase in CD62L$^-$ cells amongst CD4$^+$CXCR5$^+$PD-1$^+$ follicular T cells in autoimmune chimeras (Fig. 1g).

Although broad changes in cluster frequency were not observed, differential expression analysis revealed significant transcriptional changes between autoimmune and non-autoimmune chimeras, both amongst all follicular T cells and within individual clusters (Fig. 2a, Supplementary Data 2). Follicular T cells from autoimmune chimeras increased expression of lymphocyte-antigen 6 (Ly6a), the long noncoding RNA Gm42031, and the checkpoint receptor Lag3 and decreased expression of the transcription factor Id3 (Fig. 2b). These transcriptional changes were validated by sorting CD4$^+$CXCR5$^+$PD-1$^+$GITR$^-$ $T_{FH}$ and CD4$^+$CXCR5$^+$PD-1$^+$GITR$^+$ $T_{FR}$ cells from autoimmune and non-autoimmune chimeras and performing qRT-PCR, revealing ~5-fold upregulation of Gm42031 and ~2-fold upregulation of Ly6a (Supplementary Fig. 2). Differential expression was confirmed at the protein level by flow cytometry (Fig. 2c, d) and immunofluorescence (Supplementary Fig. 2), demonstrating increased expression of Stem Cell Antigen-1 (Sca-1, protein name for Ly6a) in follicular T cells in autoimmune chimeras. Notably, follicular T cells from wild type mice exposed to chronic foreign antigen or isolated from mesenteric lymph nodes also expressed decreased Sca-1, even after 12 weeks (Supplementary Fig. 2), suggesting that this difference is due to the autoimmune environment and not chronic exposure to antigen.

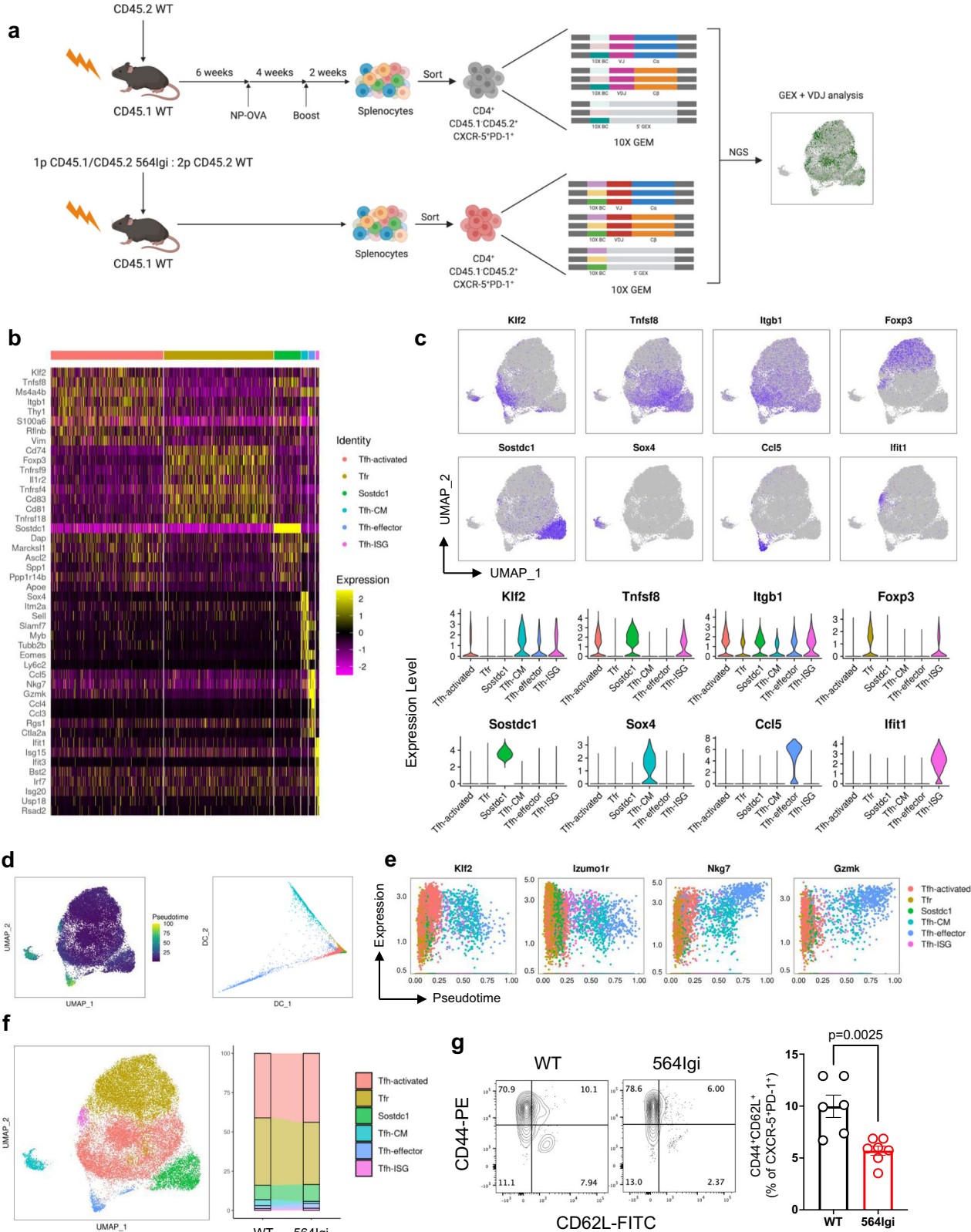

To test whether the differential expression observed in our mouse model of B cell driven autoimmune disease is recapitulated in human disease, we examined scRNA-seq data performed on CD45$^+$ leukocytes from renal biopsies from SLE patients made available through the Accelerating Medicines Partnership (AMP)[10,11]. T cells were identified by unbiased clustering of all leukocytes and unbiased clustering was performed again to identify $T_{FH}$, effector CD8, Treg, resident memory CD8, central memory CD8, and interferon stimulated CD4 cells defined by the expression of canonical markers such as *CD40LG, GZMB, FOXP3, ITGA1*, and *EOMES* (Supplementary Fig. 2). In addition to an overall increase in the number of kidney-resident T cells, SLE kidneys had an increased proportion of resident memory and central memory CD8 cells (Supplementary Fig. 2). Although

**Fig. 1 Identification of six follicular T cell clusters by scRNA-seq. a** Schematic of experimental design to generate autoimmune bone marrow chimeras and isolate follicular T cells, followed by generation of 10X Gel Bead-In Emulsions (GEM) and single cell next-generation sequencing (NGS) to generate paired gene expression (GEX) and VDJ sequences. **b** Heatmap of individual cell's (columns) expression of the top eight differentially expressed genes (rows) for each cluster (top colors). Log-normalized expression scaled for each gene. **c** Gene expression of cluster-defining genes projected onto UMAP (top) or within individual clusters (bottom) identified by scRNA-seq and unbiased clustering of follicular T cells. **d** Pseudotime projected onto UMAP (left) and diffusion map of diffusion component (DC) embeddings of cells colored by clusters (right). **e** Dot plots of correlation between pseudotime and gene expression of cells colored by cluster. **f** UMAP visualization of follicular T cells colored by unbiased cluster assignment (left) and stacked bar graph of percent of cells belonging to each cluster between wild type (WT, $n = 5$) and autoimmune (564Igi, $n = 5$) chimeras (right). **g** Flow cytometry contour plots (left) and quantification (right) of central memory frequency amongst CXCR5+PD-1+ cells from wild type (WT, $n = 6$) and autoimmune (564Igi, $n = 7$) chimeras. Data are represented as mean ± SEM. *P*-value computed using two-tailed Student's *t*-test. Source data are provided as a Source data file.

limited by sparse lymphocytic renal infiltration in healthy controls, differential expression analysis within individual clusters revealed decreased expression of *AIM1* and *IL7R* and increased expression of *KLF2* in kidney-resident $T_{FH}$ cells from SLE patients compared to healthy controls, as well as increased expression of *CX3CR1* in kidney-resident CD8 cells (Supplementary Fig. 2). We supplemented this scRNA-seq analysis with a second human dataset of bulk RNA-seq performed on CD4+ T cells isolated from peripheral blood mononuclear cells from SLE patients and healthy controls[12]. Human gene names were converted to mouse orthologs using BioMart and DEGs (absolute $\log_2 FC > 0.2$, adjusted *p*-value <0.05) were compared amongst all three datasets (Supplementary Fig. 2). Correlational analysis of fold change in gene expression between mouse and human datasets revealed that *Il7r*, *Id3*, and *Tnfsf11* are commonly decreased and *Ccl4*, *Lag3*, *Ascl2*, *Nkg7*, and *Spp1* are commonly increased in SLE and autoimmune chimeras (Supplementary Fig. 2). Patient stratification using SLE Disease Activity Index (SLEDAI) scores revealed that decreased *ID3* and increased *LAG3* expression is independent of inactive or active disease states (Supplementary Fig. 2). These data suggest that human CD4+ T cells are also transcriptionally distinct, albeit with different patterns of DEGs, in SLE.

To study the biological relevance of the DEGs observed in autoimmune chimeras, we performed gene ontology analysis on DEGs between autoimmune and non-autoimmune follicular T cells. Biological theme comparison revealed an association between hypoxia and glycolysis related gene sets and autoimmune chimera DEGs (Fig. 2e). Network visualization revealed this was due to increased expression of *Tpi1*, *Pkm*, and *Aldoa* (Fig. 2f). In contrast, follicular T cells from autoimmune chimeras decreased expression of cell adhesion related genes, including *Itgb1*, *Il7r*, and *Thy1* (Fig. 2f). Gene set enrichment analysis confirmed these associations, finding positive enrichment for a glycolytic module (M5937) and negative enrichment for cell adhesion (GO:0034113) in the ranked DEGs from follicular T cells from autoimmune chimeras (Fig. 2g). These findings suggest that follicular T cells are transcriptionally distinct in B cell driven autoimmune disease, with notable changes in metabolic state and cell–cell adhesion.

**Follicular T cell clusters are clonotypically distinct**. In parallel to scRNA-seq, we performed scTCR-seq on follicular T cells isolated from autoimmune and non-autoimmune chimeras. Of the 134,614 reads recovered, 85,955 (64%) represented full length and productive VDJ sequences, defined by an open reading frame spanning the entire VDJ sequence. These reads belonged to 43,896 unique cells, 33,833 (77%) of which had paired full length TCRα and TCRβ chains. Clonotypes were defined by identifying all cells with identical CDR3α and CDR3β amino acid sequences. After filtering, we retained 15,565 cells representing 6,745 unique clonotypes from autoimmune chimeras and 18,268 cells representing 8,871 unique clonotypes from non-autoimmune

chimeras. Repertoire-wide CDR3 length and consensus sequences (Supplementary Fig. 3) were not significantly different in autoimmune and non-autoimmune chimeras, and unbiased hierarchical clustering of variable gene usage was unable to distinguish autoimmune follicular T cells (Supplementary Fig. 3).

Unweighted network analysis of expanded clonotypes revealed limited clonotypic sharing amongst individual samples (Fig. 3a). Of the 15,608 unique clonotypes identified, only eight were identified in both autoimmune and non-autoimmune mice (Fig. 3b). Given the limited number of clonotypes shared between autoimmune and non-autoimmune mice, we first sought to examine if the TCR repertoire in its entirety could be used to distinguish autoimmune and non-autoimmune follicular T cells. We performed variable autoencoder (VAE)-based featurization of the TCR repertoires of individual chimeras to train an unbiased neural network using DeepTCR[13] (Supplementary Fig. 3). Although autoimmune repertoires are somewhat distinguishable based on Kullback–Leibler divergence (Fig. 3c), the unsupervised classification algorithm was unable to distinguish repertoires based on autoimmunity as assessed by area under the curve (AUC, Fig. 3d). UMAP visualization of TCR repertoire featurization confirmed an inability to distinguish clonotypes based on autoimmune condition (Supplementary Fig. 3). These results suggest that entire follicular T cell repertoires are not intrinsically distinct in autoimmune chimeras. However, given the presence of sparse clonotypes shared amongst individual chimeras, we next examined whether these public clonotypes might be capable of distinguishing autoimmune and non-autoimmune follicular T cells. Adapting non-metric multidimensional scaling (NMDS) analyses typically used to distinguish ecological communities using shared species, we compared similarities amongst individual chimeras using shared clonotypes, observing that public clonotypes are capable of separating chimeras on the basis of autoimmunity (Fig. 3e). This was confirmed by unbiased hierarchical clustering of public clonotypes (Fig. 3f), revealing similarity in sample distribution amongst clonotypes from autoimmune chimeras. Together these findings suggest that although entire follicular T cell repertoires are not distinguishable between autoimmune and non-autoimmune chimeras, public clonotypes are preferentially shared amongst follicular T cells from autoimmune chimeras.

To examine whether clonotypic expansion is altered in autoimmune chimeras, we compared the geometric means of clone sizes calculated as the number of individual cells identified within that clonotype. No significant differences were observed in clone size between autoimmune and non-autoimmune chimera follicular T cells (Fig. 3g). Clonal expansion within individual clusters determined by scRNA-seq was also broadly similar between autoimmune and non-autoimmune chimera follicular T cells (Fig. 3h). Clonotypes were largely restricted to individual clusters, and NMDS analysis revealed that central memory $T_{FH}$ cells were the most clonotypically distinct subset (Supplementary Fig. 3). When clonotypes were shared amongst follicular T cell

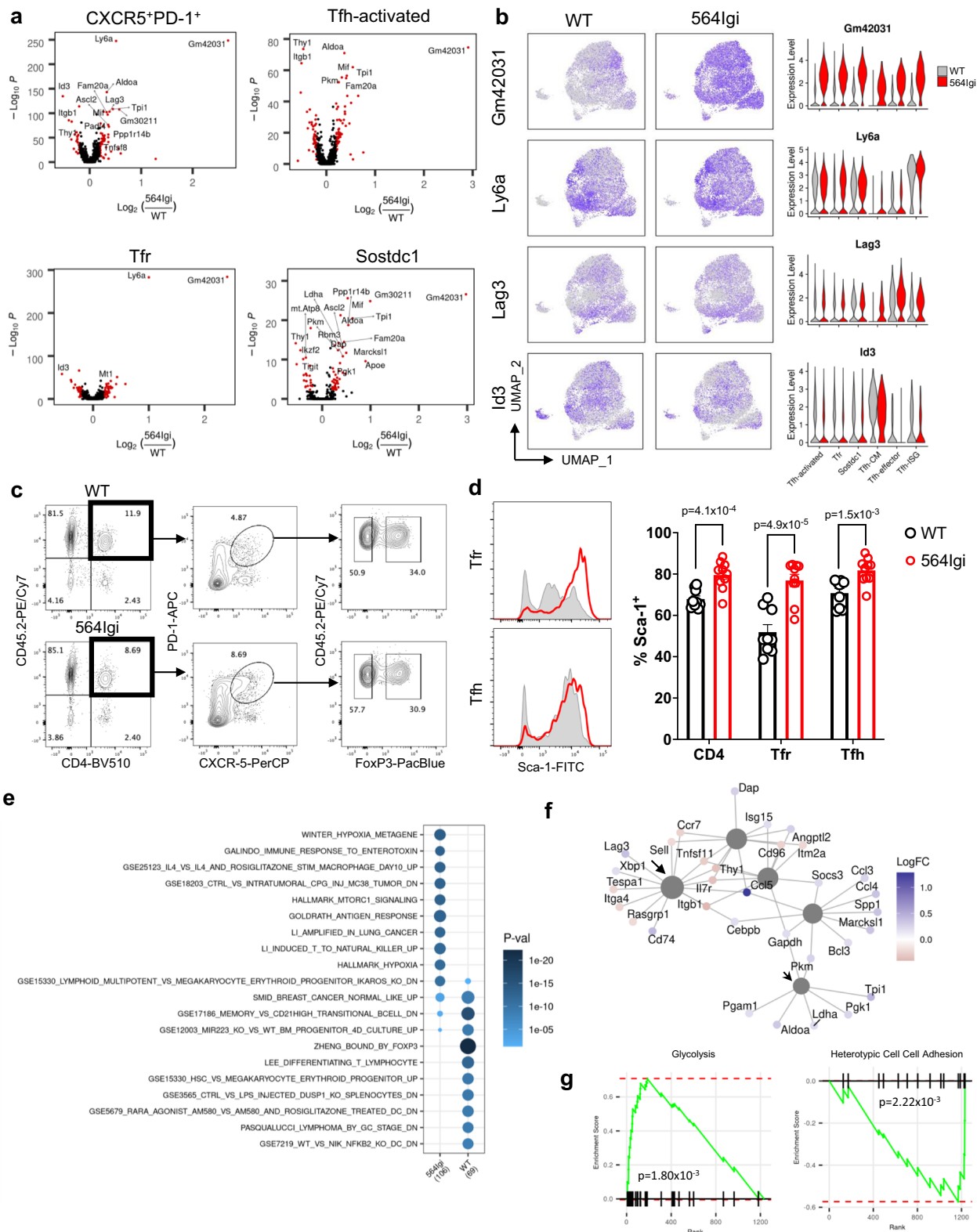

clusters they nearly always belonged to one predominant cluster, with the notable exception of clonotype sharing between activated and *Sostdc1*+ T$_{FH}$ cells (Supplementary Fig. 3). These findings suggest that in addition to their transcriptional differences, the follicular T cell clusters we identified by scRNA-seq are also clonotypically distinct. We performed rarefaction analysis to quantify diversity within each cluster of follicular T cells (Fig. 3i), observing similar levels of clonality between autoimmune and

non-autoimmune chimera follicular T cells. Together, these data suggest that follicular T cells do not have altered levels of expansion or clonality in autoimmune chimeras.

To investigate the transcriptomic differences associated with expanded clonotypes, we paired gene expression data with clonotype identity from individual cells using cell barcodes. From our filtered populations of scRNA-seq barcodes, 10,478 cells (78%) from autoimmune chimeras and 11,267 cells (74%) from

**Fig. 2 Autoimmune follicular T cells express increased Sca-1 and glycolytic genes. a** Volcano plots of differentially expressed genes between mixed autoimmune (564Igi) versus wild type (WT) chimera follicular T cells (CXCR5$^+$PD-1$^+$, top left) or within assigned clusters. Adjusted *P*-value < 0.01 and absolute log$_2$FC > 0.2 shown in red. Differential expression computed by MAST and adjusted for multiple comparison based on Bonferroni correction. **b** Expression level of select differentially expressed genes projected onto UMAP (left) or within individual clusters (right) between WT (gray) or mixed 564Igi (red) chimeras. **c** Gating strategy to identify T follicular helper (T$_{FH}$) cells and T follicular regulatory (T$_{FR}$) cells from bone marrow chimeras. **d** Flow cytometry histograms (left) and quantification (right) of Sca-1 expression in T$_{FH}$ and T$_{FR}$ cells from WT (black, $n = 9$) or mixed 564Igi (red, $n = 10$) chimeras. **e** Biological theme comparison of annotated gene sets between WT and mixed 564Igi chimera follicular T cells. Size represents gene ratio and color represents *P*-value. **f** Network plot of five most significant gene sets enriched in differentially expressed genes between mixed 564Igi versus WT chimera follicular T cells. Gray circles represent gene sets, colored dots represent genes colored by log fold change in mixed 564Igi compared to WT chimera follicular T cells. Arrow identifies leukocyte adhesion module (GO:0007159) and arrowhead identifies glycolytic module (M18792). **g** Gene set enrichment plot of indicated gene module against genes ranked by fold enrichment in mixed 564Igi chimera follicular T cells. Data are represented as mean ± SEM. *P*-value computed using two-tailed Student's *t*-test. Source data are provided as a Source data file.

WT chimeras had matching full-length and productive sequences for both TCRα and TCRβ chains. Mapping clone size onto UMAP visualization of gene expression data revealed differential distributions of clonotype expansion amongst transcriptomic space in autoimmune mice (Fig. 4a), although this was largely due to the largest clones and not generalizable amongst individual chimeras (Supplementary Fig. 4). Comparison of clonotype sizes amongst autoimmune or non-autoimmune chimeras confirmed limited clonotype sharing between chimeras and allowed us to identify clonotypes preferentially expanded in autoimmune or non-autoimmune chimeras (Fig. 4b). We observed unique CDR3 motifs and a preference for shorter CDR3β sequences (Supplementary Fig. 4) amongst clonotypes preferentially expanded in autoimmune chimeras. Clone size comparison between T$_{FR}$ and T$_{FH}$ clusters confirmed limited clonotype sharing between follicular T cell subsets, as well as differences in CDR3 motifs and length amongst clonotypes preferentially enriched in each cluster (Supplementary Fig. 4). These data suggest that although differences in CDR3 structure were not appreciated at the repertoire level, physical differences are apparent amongst condition or cluster-enriched clonotypes. Differential expression analysis of expanded clonotypes revealed similar transcriptional differences as seen amongst all follicular T cells, including increased expression of *Gm42031* and decreased expression of *Id3* and *Itgb1* amongst clonotypes expanded in autoimmune chimeras (Fig. 4c, d). These data indicate that expanded clonotypes are transcriptionally distinct in autoimmune chimeras.

We next asked whether clonal expansion itself is associated with differential gene expression. Using clone sizes determined from scTCR-seq, we correlated expression of each gene with clone size across all follicular T cells (Fig. 4e), observing known genes positively associated with clonal expansion such as *Pdcd1* ($\rho = 0.17$, $P = 2.41 \times 10^{-126}$ by Spearman) and negatively associated with clonal expansion such as *Sell* ($\rho = -0.16$, $P = 1.78 \times 10^{-122}$ by Spearman). Individual clusters had unique patterns of gene expression correlation with clonal expansion, such as negative association between *Foxp3* expression and activated T$_{FH}$ clone size, and positive associations between *Ikzf2* and *Tigit* expression and T$_{FR}$ clone size (Supplementary Fig. 4). Patterns of gene expression and clonal expansion correlation were largely similar between autoimmune and non-autoimmune chimera follicular T cells (Fig. 4f), with the notable exceptions of a greater negative correlation between *Selplg* and *Ifngr* expression and clone size in non-autoimmune follicular T cells, and a positive correlation between *Tcf7* expression and non-autoimmune follicular T cell clone size despite a negative correlation between *Tcf7* expression and autoimmune follicular T cell clone size (Fig. 4g). Amongst T$_{FR}$ cells, *Cd74* was strongly associated with clonal expansion only in autoimmune chimeras (Fig. 4f). These findings indicate that although transcriptional changes associated with clonal expansion are largely conserved,

there is a subset of genes that are distinctly positively or negatively correlated with clonal expansion in autoimmune follicular T cells.

**Predicted specificity of autoimmune follicular T cells**. To probe the antigen specificity of unknown TCRs, we applied the grouping of lymphocyte interactions by paratope hotspots (GLIPH2) algorithm[14] to our scTCR-seq data. GLIPH2 is capable of efficiently analyzing millions of CDR3 sequences and grouping them according to their predicted antigen specificity based on local motifs and global similarity (Fig. 5a). From our input of 18,517 unique clonotypes and a mouse CD4 reference dataset, GLIPH2 clustered 14,964 (81%) unique clonotypes into 6,177 specificity groups (Supplementary Data 3). To ensure high fidelity of antigen prediction, we filtered for specificity groups with at least four unique clonotypes from at least three samples with significant V-gene bias ($P < 0.05$ by GLIPH2) and significant final score ($P < 1 \times 10^{-5}$ by GLIPH2), leaving 159 specificity groups representing 1,349 unique clonotypes. These specificity groups were cross-referenced against our scRNA-seq and scTCR-seq datasets based on clonotype CDR3αβ sequences, allowing us to predict the specificity of 988 cells (7.4%) from autoimmune chimeras and 1,291 cells (8.5%) from non-autoimmune chimeras.

Specificity group membership size, determined by the total number of cells belonging to a given group, was uniformly distributed in UMAP space and between autoimmune and non-autoimmune chimeras (Supplementary Fig. 5). Surprisingly, comparison of specificity group sizes between autoimmune and non-autoimmune chimeras revealed most specificity groups were highly and equally expanded in autoimmune and non-autoimmune chimeras (Fig. 5b). Indeed, the largest specificity groups, such as GLIPH_506, were prevalent amongst both autoimmune and non-autoimmune follicular T cells (Fig. 5c). NMDS analysis was unable to distinguish autoimmune and non-autoimmune chimeras based on specificity group distributions amongst individual chimeras (Supplementary Fig. 5), reflecting the degree of antigen specificity convergence amongst follicular T cells. These findings suggest that most follicular T cells share antigen specificities in autoimmune and non-autoimmune chimeras. Specificity groups were also highly concordant between T$_{FR}$ and T$_{FH}$ clusters, and the specificity groups included clonotypes from distinct clusters with few specificity groups observed in only one cluster (Supplementary Fig. 5). These results suggest that although T$_{FR}$ and T$_{FH}$ cells might be clonotypically distinct, they are likely capable of binding similar antigens.

Despite the prevalence of specificity groups sharing between autoimmune and non-autoimmune chimeras, some specificity groups were preferentially expanded in a given condition, such as GLIPH_4117 and GLIPH_4448 (Fig. 5b, c). CDR3 motif analysis identified unique motifs in both shared and condition-specific specificity groups (Fig. 5d) that are predicted to be contact

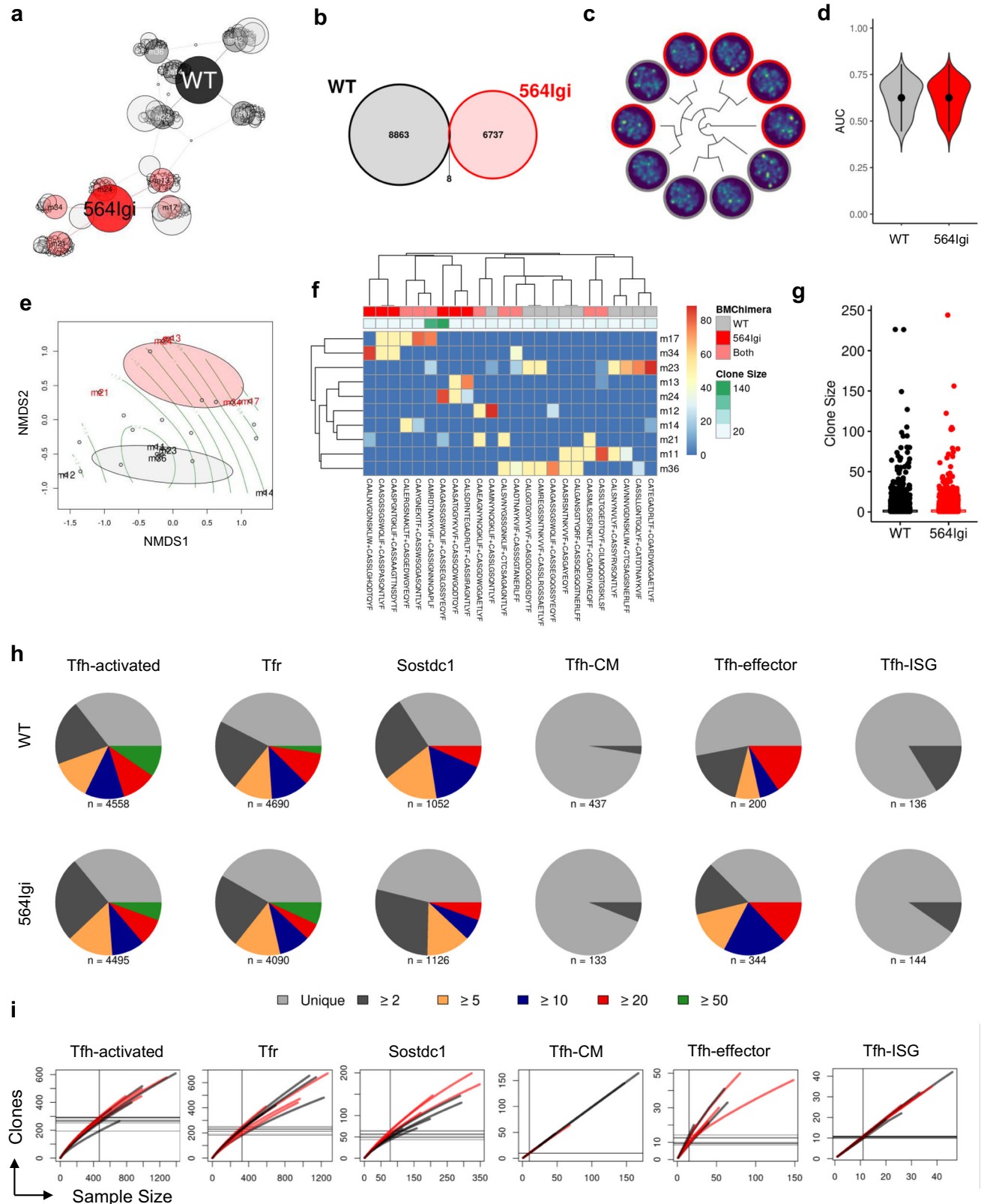

residues by antigen-specific paratope convergence[15]. To better understand the clonotypic composition and relationships between these shared and condition-specific specificity groups, we performed unweighted network analysis of the largest specificity groups and specificity groups preferentially expanded in auto-immune or non-autoimmune chimeras (Fig. 5e). Public

specificity groups (absolute $\log_2$FC < 2.5) were often polyclonal with limited clonotype sharing amongst specificity groups. In contrast, condition-specific specificity (absolute $\log_2$FC > 2.5) groups were pauciclonal, sharing clonotypes with public specificity groups but not with each other (Fig. 5b, e). These findings suggest that non-shared specificity groups represent pauciclonal

**Fig. 3 Public clonotypes are distinct between autoimmune and non-autoimmune follicular T cells. a** Unweighted network analysis of expanded clonotypes (>10 individual cells) from WT (black) and mixed 564Igi (red) chimeras. Clonotypes are defined by paired full length TCRα and TCRβ sequences. Individual samples are depicted as colored circles, clonotypes are depicted as gray circles and sized according to number of cells belonging to given clonotype. Edges represent clonotype membership to individual samples. **b** Venn diagram comparing number of unique and shared clonotypes between WT (black) and mixed 564Igi (red) chimeras. **c** Repertoire dendrogram of variable autoencoder-based featurization of TCR repertoires of individual samples represented in UMAP space. Dendrogram represents inter-repertoire symmetric Kullback–Leibler divergence computed from sample-agnostic network-based clustering. Color of node indicates sample condition (gray, WT; red, 564Igi). **d** Performance of K-nearest-neighbors instance-based unsupervised classification algorithm applied to TCR repertoire featurization assessed by area under the curve (AUC). Data are represented as mean ± SD. **e** Ordinance plot of non-metric multidimensional scaling (NMDS) of clonotype distribution amongst individual samples. Text indicates sample name and color indicates sample condition (gray, WT; red, 564Igi). Individual clonotypes are depicted as gray circles. **f** Hierarchical clustering of public clonotypes based on distribution amongst chimeras (rows). Column colors indicate sample condition of given clonotype (gray, WT; red, 564Igi; light red, both) and number of individual cells belonging to each clonotype. Columns are labeled with CDR3 sequences from TCRαβ pairings. **g** Number of individual cells belonging to each clonotype amongst WT (black) or mixed 564Igi (red) chimeras. **h** Pie charts of clonal expansion of follicular T cell clusters identified by scRNA-seq (columns) in WT (top) or mixed 564Igi (bottom) chimeras. Number of cells with both TCRα and TCRβ successfully identified is shown below each pie chart. For clonotypes expressed by two or more cells, the number of cells expressing that clone is shown by a distinct color. **i** Rarefaction curves of clonotype richness within follicular T cell clusters identified by scRNA-seq. Individual samples are depicted by lines and colored by sample condition (black, WT; red, 564Igi).

responses that might display limited cross-reactivity with the dominant antigen-specificities observed amongst most follicular T cells. To test this hypothesis, we labeled specificity groups based on preferential expansion (absolute $\log_2 FC > 2.5$ and total size >10) to identify condition-specific specificities (Supplementary Fig. 5). Condition-specific specificity groups had greater average clone sizes compared to shared specificity groups (Fig. 5f), consistent with pauciclonal expansion in condition-specific but not shared specificity groups. Interestingly, pairing GLIPH analysis with scRNA-seq cluster determination revealed that autoimmune-specific specificity groups consisted of a greater proportion of $T_{FR}$ cells than non-autoimmune-specific specificity groups (Fig. 5f). Rarefaction analysis of all specificity groups within individual clusters revealed similar levels of diversity between autoimmune and non-autoimmune chimera follicular T cells (Supplementary Fig. 5), reinforcing our earlier finding that follicular T cells do not have altered levels of expansion or clonality in autoimmune chimeras. Together these results suggest that most of the follicular T cell repertoire is unexpanded yet shares antigen reactivities, but in autoimmune or non-autoimmune conditions a subset of the repertoire participates in similar degrees of pauciclonal expansion to react to unique antigens.

**Follicular T cell specificity groups are transcriptionally distinct.** Given the phenotypic differences observed between shared and condition-specific specificity groups, we next asked whether specificity groups are transcriptionally distinct in autoimmune disease. Differential expression analysis of condition-specific specificity groups (Fig. 6a) revealed increased expression of *Rln3*, *Itgb7*, and *Gm42031* and decreased expression of *Tcf7*, *Nsg2*, and *Asap1* in cells belonging to specificity groups preferentially expanded in autoimmune chimeras (Fig. 6b, c). Gene ontology analysis demonstrated the importance of these genes in lymphocyte differentiation, activation, and adhesion (Fig. 6d), suggesting that cells undergoing pauciclonal expansion in autoimmune chimeras towards shared antigens increase expression of genes related to germinal center trafficking and entry. Biological theme comparison between shared and non-shared specificity groups (Fig. 6e) revealed an association between non-autoimmune enriched specificity groups and naïve (GSE20366) gene sets, whereas autoimmune enriched specificity groups were associated with the interferon gamma response (M5913). To pair these gene expression profiles with antigen specificity, the average expression of all genes amongst all cells within each individual specificity group was used to calculate a module score for

lymphocyte differentiation (GO:0030098) and T cell activation (GO:0042110). Comparison of specificity group size between autoimmune and non-autoimmune mice coupled with module scores or gene expression confirmed that cells belonging to specificity groups enriched in autoimmune chimeras increase expression of differentiation genes and *Itgb7*, whereas cells belonging to shared specificity groups decrease expression of differentiation genes (Fig. 6f).

Differential expression of T cell activation and adhesion related genes between autoimmune and non-autoimmune specificity groups led us to hypothesize that follicular T cells with autoimmune enriched specificities have altered germinal center entry and trafficking. Indeed, we observed a ~2.2-fold increase in PSGL-1$^{lo}$CD62L$^{lo}$ extrafollicular (EFO) CD4$^+$ T cells in autoimmune chimeras (Supplementary Fig. 6). Increased expression of PD-1 amongst EFO CD4$^+$ T cells in autoimmune chimeras (Supplementary Fig. 6) coupled with the positive association between *Selplg* expression and autoimmune enriched follicular T cell specificities (Fig. 6c) suggests that EFO CD4$^+$ T cells bind shared rather than autoimmune enriched antigens. Together, these results suggest that follicular T cells undergoing pauciclonal expansion towards shared antigens modulate expression of germinal center trafficking genes in autoimmune disease.

As each specificity group consists of multiple clonotypes with varying levels of expansion (Supplementary Fig. 6), we next examined whether clonal expansion within specificity groups is associated with differential gene expression. Clonal expansion associated gene expression was markedly distinct between autoimmune and non-autoimmune enriched specificity groups (Supplementary Fig. 6), suggesting that expansion against different antigens can drive differential gene expression. Clonal expansion was associated with increased expression of *Malat1*, *Hcst*, and *Ptprc* amongst cells with specificities enriched in non-autoimmune chimeras despite a negative association amongst cells with specificities enriched in autoimmune chimeras (Fig. 6e). To test whether the differences we observed between cells belonging to autoimmune or non-autoimmune enriched specificity groups are recapitulated within a single specificity group, we examined GLIPH_506, one of the largest shared specificity groups between autoimmune and non-autoimmune chimeras (Supplementary Fig. 6). Follicular T cells belonging to this specificity group from autoimmune chimeras expressed increased cytotoxic effector genes and *Gm42031* and *Stat1*, and decreased *Cd4* (Supplementary Fig. 6). These findings suggest that even amongst follicular T cells with identical predicted specificities, autoimmunity results in differential gene expression.

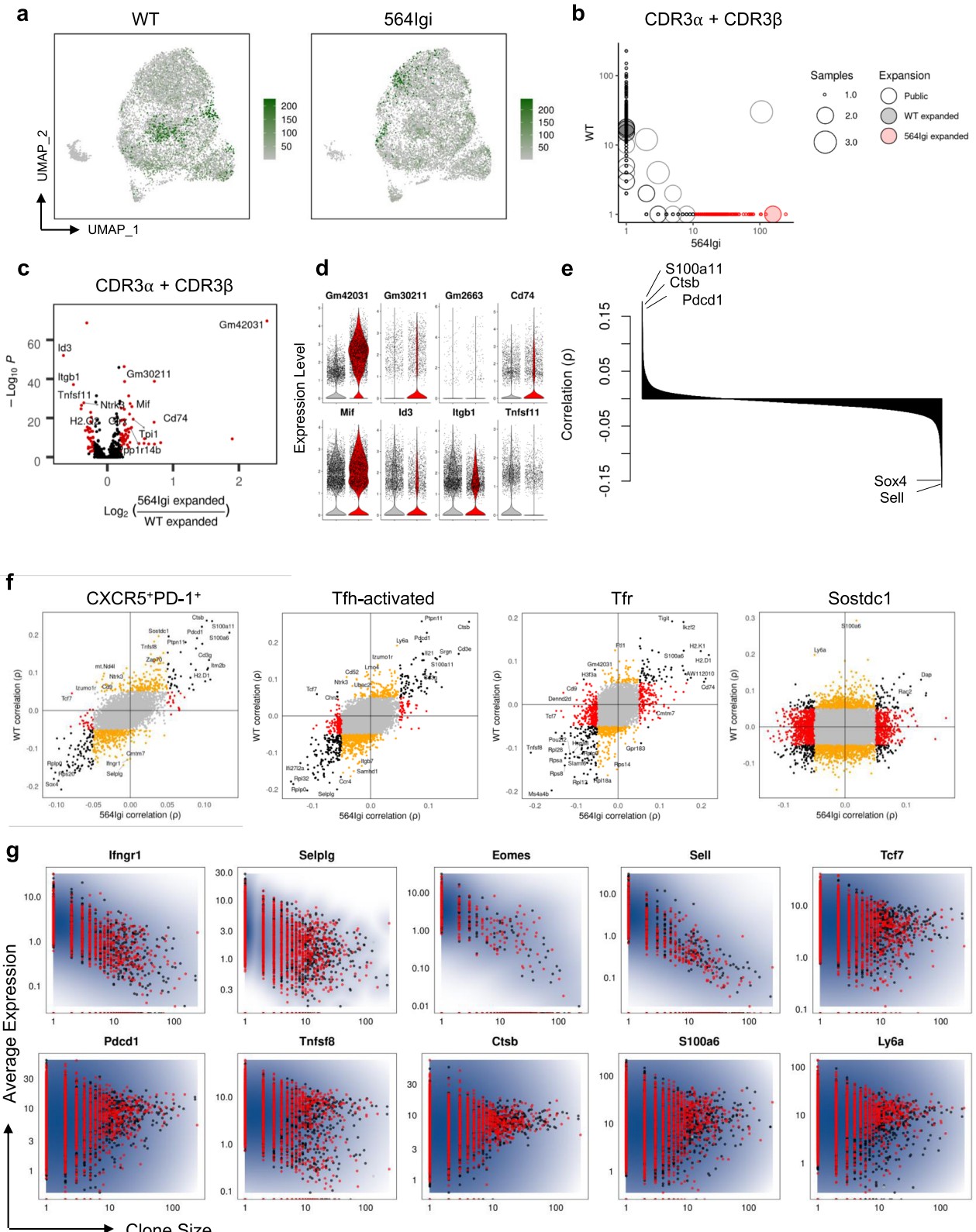

**Machine learning predicts follicular T cell reactivities**. To reveal the antigen specificity of unknown TCR sequences, we queried the VDJdb[16], PIRD[17], and McPAS-TCR[18] databases of known CDR3β specificities. We compiled all three databases to create a single reference database consisting of 99,809 unique CDR3β sequences with annotated specificities for 469 antigens and 482 peptides from 105 different diseases or experiments

(Supplementary Data 4). This database was first used to search for matching CDR3β sequences between known TCR specificities and our scTCR-seq data. Amongst the 13,232 unique CDR3β sequences identified from follicular T cells, 296 (2.23%) matched CDR3β sequences present in our annotated reference database with a collective specificity for 61 unique peptides (Supplementary Fig. 7). Successfully annotated CDR3β sequences represented

**Fig. 4 Distinct genes are associated with clonal expansion in autoimmune follicular T cells. a** Clone size mapped onto UMAP visualization of transcriptomic data of individual follicular T cells from WT (left) or mixed 564Igi (right) mice. Clonotypes are defined by paired full length TCRα and TCRβ sequences and clone sizes are number of individual cells within a given clonotype. **b** Scatter plot comparing clone size between WT and mixed 564Igi chimeras. Clonotypes are colored according to preferential expansion (absolute $\log_2$FC > 3 and clone size >10) in WT (black) or mixed 564Igi (red) chimeras and sized according to number of samples in which the given clonotype is observed. **c** Volcano plots of differentially expressed genes between cells belonging to clonotypes preferentially expanded in mixed 564Igi versus WT chimeras indicated in (**b**). Adjusted $P$-value <0.01 and $\log_2$FC > 0.2 shown in red. Differential expression computed by MAST and adjusted for multiple comparison based on Bonferroni correction. **d** Expression of indicated genes in cells belonging to preferentially expanded clonotypes in WT (gray) or mixed 564Igi (red) chimeras. **e** Rank order plot of Spearman's correlation coefficients of each gene with clone size across all cells. **f** Scatter plots comparing correlation coefficients of each gene with clone size between WT and mixed 564Igi chimeras within assigned clusters. Correlation coefficients >0.05 are indicated in black (correlated in both conditions), red (correlated in mixed 564Igi chimeras only), or red (correlated in WT chimeras only). **g** Average expression of indicated gene amongst all cells belonging to individual clonotypes versus clone size. Clonotype color indicates sample condition (black, WT; red, 564Igi).

both public and private CDR3β sequences amongst autoimmune and non-autoimmune chimeras, although the most expanded CDR3β clones were not matched (Supplementary Fig. 7). The annotated peptides associated with known CDR3β sequences preferentially expanded in each condition were used to create a positional-weighted matrix (PWM) of amino acid preference scored by clone size and fold-enrichment within a given condition (Supplementary Fig. 7). To predict novel peptide specificities, we used these PWMs to perform profile-based scoring of the mouse proteome using a 9-mer sliding profile-based search. Our profile-based search of the mouse proteome yielded 1,071 unique peptides from 1,176 unique antigens using the non-autoimmune PWM and 47,636 unique peptides from 16,970 unique antigens using the autoimmune PWM (Supplementary Fig. 7). Notably, candidate antigens for non-autoimmune follicular T cells were nearly entirely shared with autoimmune follicular T cells.

To extend our antigen prediction methodology to computational specificity group predictions, GLIPH2 clustered 19,290 (19%) unique CDR3β sequences from our reference database of known antigen specificities into 13,695 specificity groups. CDR3β local-motif patterns identified by GLIPH2 were cross-referenced to annotate our follicular T cell specificity groups with predicted antigen specificities extracted from reference databases (Fig. 7a). Database annotation succeeded in predicting antigen specificities for 39/159 (25%) of follicular T cell specificity groups, predicting autoimmune, parasite, cancer, or viral related antigen reactivity amongst specificity groups shared between autoimmune and non-autoimmune chimeras, and viral antigen reactivity amongst non-autoimmune enriched specificity groups (Fig. 7b). Specificity groups with predicted reactivity for autoimmune related antigens consisted of clonotypes identified from both autoimmune and non-autoimmune chimeras (Fig. 7b), possibly reflecting that even in non-autoimmune chimeras follicular T cells are capable of binding self-antigens. Annotation of UMAP visualization with antigen and peptide predictions revealed both local (FIDCYLLAI) and widespread (NLVPMVATV) distributions of predicted antigen-specific cells in transcriptional space (Fig. 7c). Together these data suggest that autoimmune and non-autoimmune follicular T cells both share reactivities against known foreign and self-antigens, as well as have condition-specific antigen enrichment.

To validate these GLIPH2-based findings, we designed a machine learning approach to predict antigen specificities. Peptides with known binding to at least 100 unique CDR3β sequences from our reference database were used for VAE-based featurization and supervised deep learning using DeepTCR[13]. TCR specificity classification was trained and validated using a Monte-Carlo cross-validation algorithm provided with the CDR3β sequences and VDJ gene usage of 53,274 unique clonotypes representing 57 unique peptides, achieving an average AUC of 0.86 (Fig. 7d). Visualization of VAE-encoded feature

space demonstrated the ability to cluster peptides by unbiased hierarchy based on TCR featurization alone (Supplementary Fig. 7). Thus, supervised TCR classification can predict antigen specificity with high sensitivity and specificity. UMAP visualization of clonotype featurization of both training and validation sets confirmed separation of peptide-specific clonotypes in feature-space (Fig. 7e). This machine learning algorithm was then applied to our unknown follicular T cell clonotypes, allowing us to represent each clonotype in our supervised classification feature-space. Clonotypes from autoimmune and non-autoimmune follicular T cells clustered together (Fig. 7f), reinforcing GLIPH2-based prediction of shared antigen specificities amongst autoimmune and non-autoimmune follicular T cells. Notably, follicular T cells from autoimmune and non-autoimmune chimeras clustered separately from any region in UMAP space annotated by our supervised classifier (Fig. 7f), suggesting that the set of possible antigens they respond to is less diverse compared to the annotated antigens present in our training dataset.

These computational methods provide a multipronged toolkit for antigen prediction from paired scRNA-seq and scTCR-seq data that together ultimately suggest that follicular T cells from both autoimmune chimeras and non-autoimmune chimeras are likely to respond to a similar set of antigens.

## Discussion

We have used paired scRNA-seq and scTCR-seq to observe that despite convergence toward shared predicted antigen specificities, follicular T cells from autoimmune and non-autoimmune chimeras remain transcriptionally distinct. We were able to identify known subsets of follicular T cells, such as $T_{FR}$, activated $T_{FH}$, and $Sostdc1^+$ $T_{FH}$ cells, as well as previously uncharacterized subsets, such as interferon-responsive $T_{FH}$ cells. $Sostdc1^+$ $T_{FH}$ cells are a recently characterized subpopulation that appear to be involved in Wnt-mediated $T_{FR}$ differentiation[19]. Patients with SLE, myasthenia gravis, Sjogren's syndrome, and rheumatoid arthritis have increased numbers of $T_{FH}$ cells[20–31], and altered $T_{FR}$ to $T_{FH}$ cell ratios have been observed in mouse models of arthritis and patients with ankylosing spondylitis, Sjogren's syndrome, myasthenia gravis, and multiple sclerosis[32–36]. Surprisingly we observed no changes in the relative proportion of $T_{FR}$, activated $T_{FH}$, or $Sostdc1+$ $T_{FH}$ cells in our mixed autoimmune chimeras, although a central memory-like $CD44^+CD62L^+$ $T_{FH}$ population was decreased. We show that this might be due to a relative increase in $PSGL-1^{lo}CD62L^{lo}$ extrafollicular $CD4^+$ T cells, which others have also observed in mouse models of autoantibody-mediated disease[37–39]. Notably, the follicular T cells described here might also represent $T_{FH}$ and $T_{FR}$ cells that are not germinal center-experienced, such as circulating[40], pre-follicular[41,42], or extrafollicular[43] $T_{FH}$ cells. Functional differences within $T_{FR}$, $T_{FH}$, and $Sostdc1^+$ $T_{FH}$ cells, rather than differences in the relative

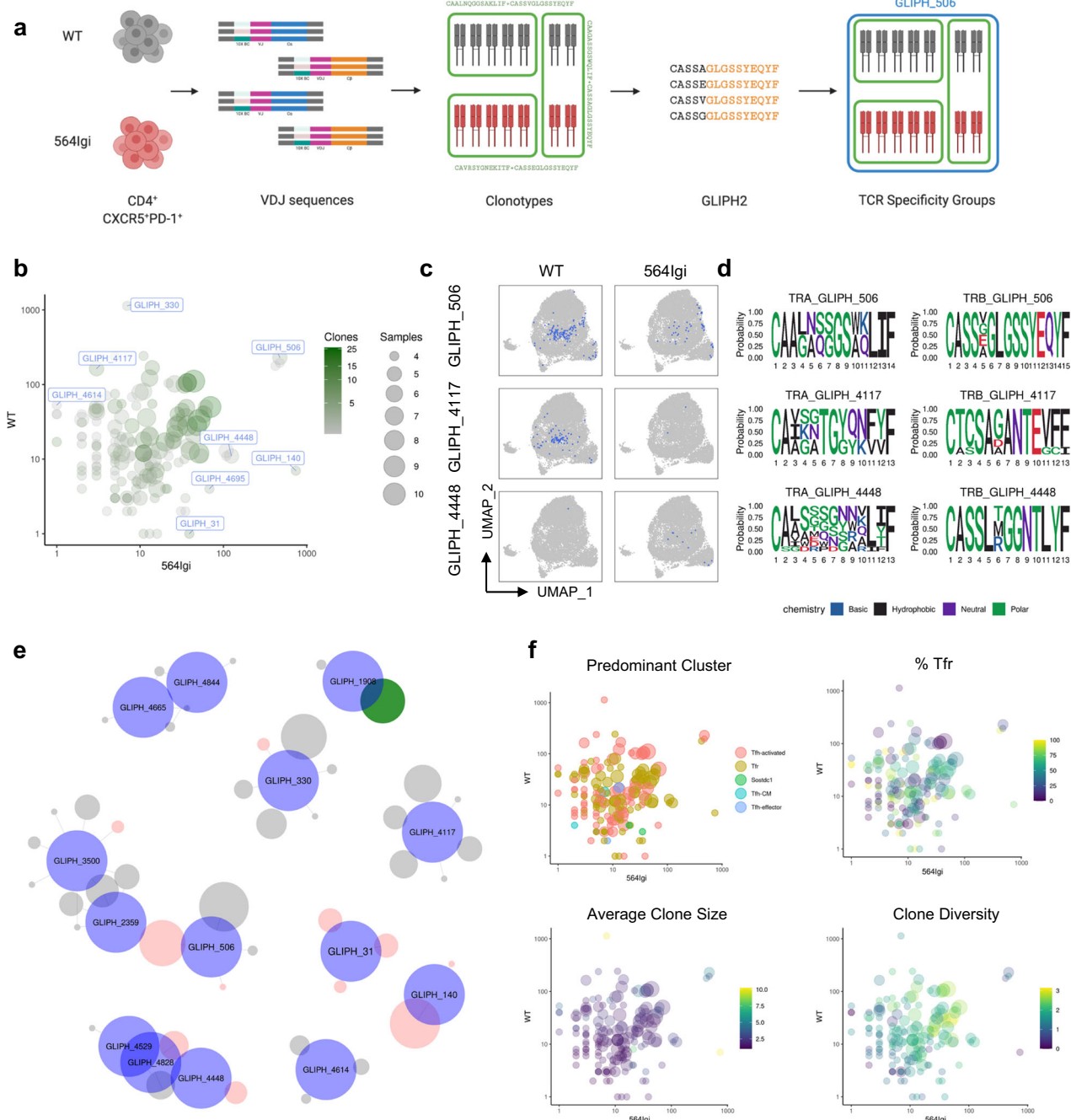

**Fig. 5 GLIPH prediction of TCR specificity shows convergence toward shared specificities in autoimmune and non-autoimmune follicular T cells. a** Schematic of computational pipeline to assign clonotype groups from scRNA-seq data, followed by assigning clonotypes to TCR specificity groups using the GLIPH2 algorithm. **b** Scatter plot comparing specificity group size between WT and mixed 564Igi chimeras. Specificity groups are colored according to the number of unique clonotypes that belong to a given specificity group and sized according to number of samples in which the given specificity group is observed. Select specificity groups are labeled with their arbitrary name in blue. **c** Mappings of indicated specificity groups onto UMAP visualization of transcriptomic data from WT (left) or mixed 564Igi (right) chimeras. **d** Motif analysis of 14 amino acid long CDR3s of TCRα (left) and TCRβ (right) chains of all cells belonging to indicated specificity group. **e** Unweighted network analysis of clonotype assignment to indicated specificity groups. Specificity groups are depicted as blue circles, clonotypes are colored according to condition (black, WT; red, 564Igi; green, both) and sized according to number of cells belonging to given clonotype. **f** Scatter plot comparing TCR specificity group size between WT and mixed 564Igi chimeras and colored according to most prevalent cluster amongst cells belonging to given specificity group (top left), percentage of cells within given specificity group assigned to the $T_{FR}$ cluster (top right), geometric mean of clone sizes of clonotypes within given specificity group (bottom left), and Shannon diversity index of clonotype expansion within given specificity group (bottom right).

frequencies of these subsets, are therefore likely responsible for the autoimmunity observed in our mixed chimera model of autoantibody-mediated disease.

Characterization of what makes a follicular T cell dysfunctional remains unclear. In our bone marrow chimera model, germinal center dysfunction can be induced by autoreactive B cells in an otherwise wild type environment, and this dysfunction begets further loss of tolerance and autoantibody development. We observed that $T_{FR}$, $T_{FH}$, and *Sostdc1*⁺ cells from autoimmune chimeras consistently express increased *Ly6a* and *Gm42031* and

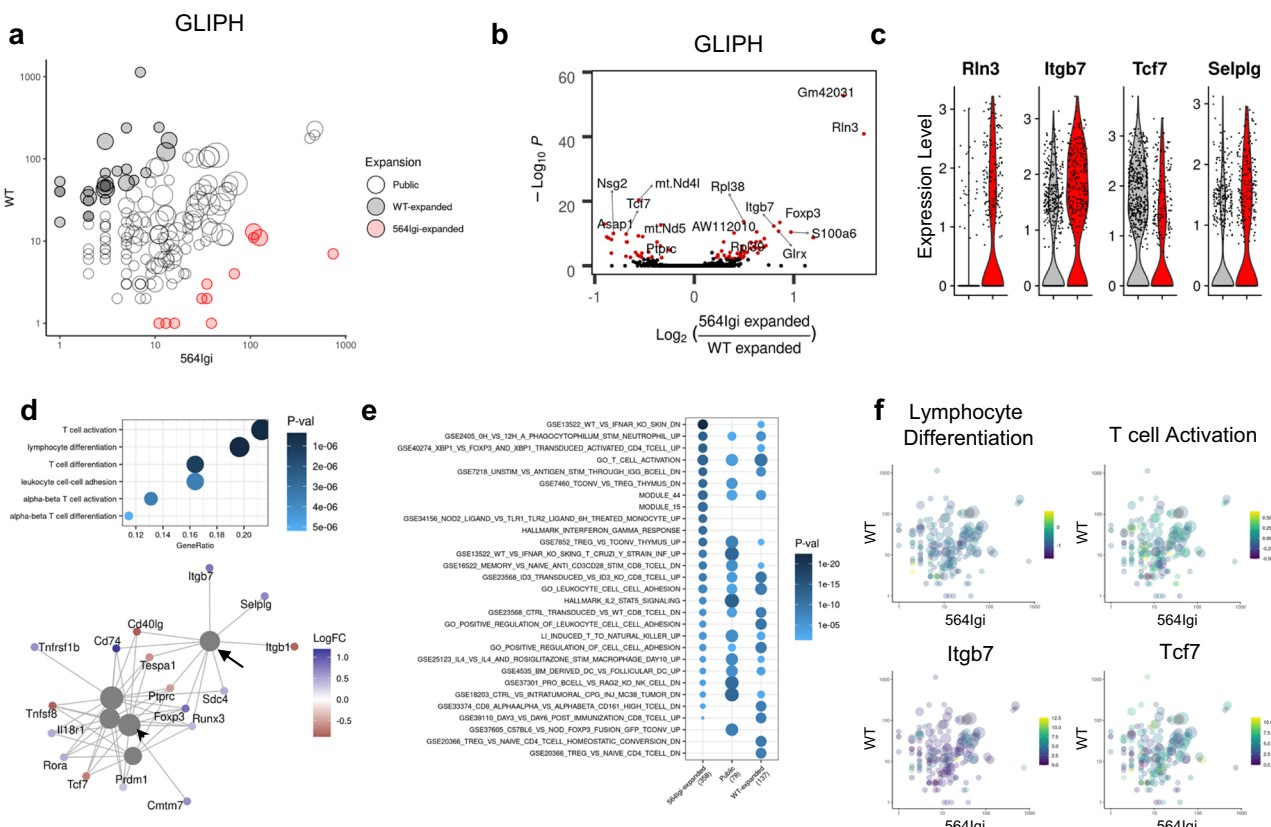

**Fig. 6 TCR specificities enriched in autoimmune disease are transcriptionally distinct. a** Scatter plot comparing specificity group size between WT and mixed 564Igi chimeras. Specificity groups are colored according to preferential expansion (absolute log$_2$FC > 2.5 and total size >10) in WT (black) or mixed 564Igi (red) chimeras and sized according to number of samples in which the given clonotype is observed. **b** Volcano plots of differentially expressed genes between cells belonging to preferentially expanded specificity groups in mixed 564Igi versus WT chimeras indicated in (**a**). Adjusted *P*-value <0.01 and log$_2$FC > 0.2 shown in red. Differential expression computed by MAST and adjusted for multiple comparison based on Bonferroni correction. **c** Violin plots of expression of indicated genes in cells belonging to preferentially expanded specificity groups in WT (gray) or mixed 564Igi (red) chimeras. **d** Gene ontology analysis (top) and network plot (bottom) of differentially expressed genes between cells belonging to preferentially expanded specificity groups in mixed 564Igi versus WT chimeras indicated in (**a**). Top; size represents gene ratio and color represents *P*-value. Bottom; gray circles represent gene sets and colored dots represent genes colored by log$_2$FC in mixed 564Igi compared to WT chimera follicular T cells. Arrow identifies leukocyte cell–cell adhesion module (GO:0007159) and arrowhead identifies lymphocyte differentiation module (GO:0030098). *P*-value computed by gene ontology enrichment analysis and adjusted for multiple comparisons using Benjamini–Hochberg procedure. **e** Biological theme comparison of annotated gene sets between cells belonging to specificity groups without a condition preference (public) or preferentially expanded in WT (black) or mixed 564Igi (red) chimeras. Size represents gene ratio and color represents *P*-value. *P*-value computed by gene ontology enrichment analysis and adjusted for multiple comparisons using Benjamini–Hochberg procedure. **f** Scatter plot comparing specificity group size between WT and mixed 564Igi chimeras and colored according to lymphocyte differentiation module score (GO:0030098, top left), T cell activation module score (GO:0042110, top right), and average expression of *Itgb7* (bottom left) and *Tcf7* (bottom right). Module scores were calculated using average expression of all genes of cells belonging to each specificity group.

decreased *Id3* and *Lag3*. *Ly6a* is a type I interferon-responsive gene fundamental for hematopoietic progenitor homeostasis[44–48], although its function on mature lymphocyte populations such as follicular T cells remains unclear. Upregulation of Sca-1 (protein name of *Ly6a*) on T and B cells is observed in the *Dnase1l3*[−/−] model of type I IFN-driven autoantibody disease[39,49], suggesting that type I IFN signaling might be mediating follicular T cell gene expression changes in our chimeric model of autoantibody-mediated disease. *Gm42031* is a long non-coding RNA (lncRNA) of unknown significance. LncRNAs are thought to modulate widespread but low-level gene expression patterns governed by transcriptional regulators by altering translation or serving as miRNA sponges[50–52]. Therefore, *Gm42031* might be responsible for enacting disparate and subtle phenotypic changes in follicular T cells, such as the alterations in glycolysis pathways we observed by gene ontology analysis and by others in the B6.*Sle1.Sle2.Sle3* model of autoantibody-mediated disease[53,54]. *Id3* is a helix-loop-

helix (HLH) protein that is critical for Treg cell maintenance and T$_{FR}$ maturation[55,56], suggesting that T$_{FR}$ developmental progression might be impaired in our chimeric model of autoantibody-mediated disease. We also observed differential associations between *Tcf7, Selplg*, and *Cd74* expression and clonal expansion in autoimmune and non-autoimmune chimeras, suggesting that T$_{FH}$ differentiation, trafficking, and macrophage inhibitory factor (MIF) response are altered in an autoimmune environment[57–60], respectively. These transcriptional differences might represent follicular T cell phenotypic changes that are permissive to the autoimmunity observed in our model, might be consequences of an already autoimmune environment, or might reflect differences in immune response kinetics between autoimmune and immunized mice. Future experiments are necessary to clarify these hypotheses and examine the functional consequences of these genes in follicular T cells. Understanding how an autoreactive B cell clone might induce these or other

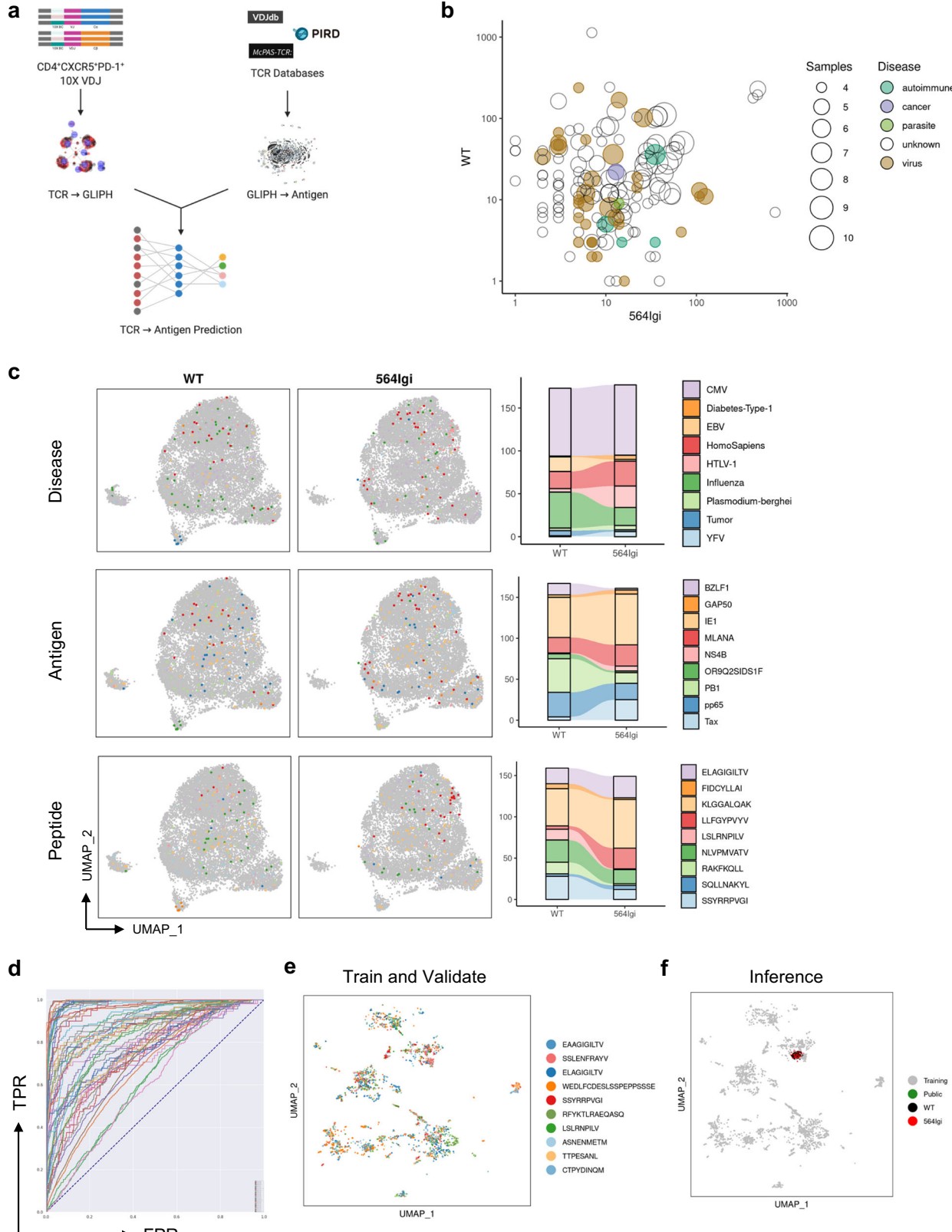

phenotypic changes in previously normal $T_{FH}$ cells would reveal a fundamental step in the march towards loss of peripheral tolerance and autoantibody development.

In addition to transcriptional differences, $T_{FH}$ cell dysfunction might be associated with changes in the TCR repertoire. Patients with mutations in *CD40LG* or that lack MHC class II expression have elevated levels of autoreactive B cells[61], suggesting that CD40 and TCR signals serve as checkpoints to maintain peripheral B cell tolerance. B cell specific deletion of MHC class II in MRL/*lpr* mice decreased autoantibody production and $T_{FH}$ cell number[62]. Recent studies using HEL3X bone marrow chimeras have demonstrated that clonal redemption of anergic autoreactive B cells requires high

**Fig. 7 CDR3β database and machine learning-based peptide prediction indicate similar reactivities in autoimmune and non-autoimmune follicular T cells. a** Schematic of computational approach to predict antigen specificity by performing GLIPH analysis on TCR databases of annotated CDR3 sequences (VDJdb, PIRD, McPAS-TCR), followed by matching follicular T cell clonotypes identified by scRNA-seq to annotated specificity groups via shared CDR3β sequences. **b** Scatter plot comparing specificity group size between WT and mixed 564Igi chimeras and colored according to disease class of predicted antigen. Size of specificity groups represents number of samples in which the given specificity group is observed. **c** Mapping of predicted disease category (top), antigen (middle), or peptide (bottom) onto UMAP visualization of transcriptomic data from WT or mixed 564Igichimeras (left) and stacked bar graph of number of cells belonging to each prediction between WT and mixed 564Igi chimeras (right). Only the nine most frequent antigens and peptides are labeled on right. **d** Receiver operator curve to assess ability of a supervised CDR3β sequence Monte-Carlo cross-validation algorithm to classify TCR specificity using annotated TCR databases for training. Classification performance assessed by area under the curve (AUC). **e** UMAP visualization of variable autoencoder based supervised CDR3β sequence featurization of annotated clonotypes from TCR databases following training and validation by a Monte-Carlo cross-validation algorithm. Clonotypes are colored according to annotated peptide (ground truth) and only peptides with AUC > 0.9 were included. Ten most common peptides are labeled on right. **f** Superimposition of supervised classification algorithm from (**e**) applied to CDR3β sequences from scRNA-seq data onto UMAP visualization of peptide specificities from the training and validation datasets. Clonotypes are colored according to condition (green, both; black, WT; red, 564Igi).

density membrane-bound foreign antigen for productive T cell collaboration, leading to rapid mutation away from self toward foreign reactivity[63]. Transgenic mouse models have demonstrated that autoreactive B cells can present self-antigen to T cells, breaking tolerance and amplifying autoantibody production independent of TLR adjuvanticity[64–66]. However, uncertainty exists over whether these CD4+ T cells represent T_FH cells or an extrafollicular T cell population[37,43,67]. Although we did not capture extrafollicular T cells in our single cell profiling, we use computational prediction to show that follicular T cells likely possess an autoreactive repertoire that can participate in these interactions. Recent studies have found that both T_FH and T_FR cells have remarkably diverse but non-overlapping repertoires, and that immunization did not significantly alter their clonality but rather resulted in non-specific T_FH bystander activation possibly due to TCR cross-reactivity[68]. These findings are consistent with our observation of infrequent clonotype sharing between clusters. We hypothesize that the surprising convergence we observed in predicted specificity groups between autoimmune and non-autoimmune mice reflects bystander activation due to the highly cross-reactive nature of the TCR[69,70], particularly given the high diversity and small clone size observed in public specificity groups. It follows that just as dysfunctional T_FH cells might help autoreactive B cells overcome clonal anergy, autoreactive B cells might help autoreactive T_FH cells overcome T cell ignorance.

Whether the T_FH cells expanded in autoantibody-mediated diseases are autoreactive and share specificities with autoreactive B cells remains unknown. Although autoreactive CD4+ T cells have been identified in autoantibody-mediated disease and correlate with disease severity[71–76], unbiased profiling of CD4+ T cell specificities in autoantibody-mediated disease has not been previously performed. Our observed convergence in predicted specificity might either reflect emergence of autoreactivity following irradiation and immunization with foreign antigen in the non-autoimmune chimeras, or bystander activation of non-autoreactive follicular T cells in the autoimmune chimeras[77–79]. Notably, interrogation of the minority of specificity groups enriched in either condition combined with computational antigen prediction identified possible foreign reactivities for non-autoimmune enriched groups only. Pairing gene expression data with specificity group prediction revealed that autoimmune enriched specificity groups modulate their expression of germinal center trafficking genes such as *Itgb7* and *Selplg*, and that cells within identical predicted antigen specificity groups had differential gene expression in autoimmune and non-autoimmune mice. Therefore, an autoreactive TCR might influence T cell trafficking, whereas an autoimmune environment might influence T cell gene expression independent of antigen specificity. Furthermore, our analyses do not address the likely varying affinities

of the many clonotypes within individual specificity groups, which possibly contributes to or correlates with gene expression variability. Further validation and peptide binding profiling are necessary to confirm and discover the predicted autoreactivity and overall antigen-specificity of the candidate TCRs and specificity groups we have generated in silico.

Here we present substantial transcriptional and clonotypic differences in the follicular T cell compartment in a mouse model of autoantibody-mediated disease. Implication of potential targets such as lncRNAs, glycolysis, and germinal center trafficking highlights pathways to prioritize for future therapeutic consideration for B cell driven diseases. Identification of cellular processes responsible for T_FH cell dysfunction not only improves our understanding of autoimmune disease pathogenesis, but also establishes potential modulators of B cell anergy and clonal redemption that might be manipulated to optimize antibody responses to foreign antigens. While clearly involved in autoantibody-mediated disease, germinal center evolution is a fundamental feature of adaptive immunity that has the potential to impact the many other diseases impacted by insufficient or overactive germinal center responses, such as vaccine design, anti-HIV broadly neutralizing antibody development, and molecular mimicry.

## Methods

**Mice.** C57BL/6J and B6.SJL (CD45.1) mice were obtained from Jackson Laboratories. 564Igi mice on a C57BL/6 background[9] were originally provided by Theresa Imanishi-Kari (Tufts University) and were maintained in-house. All mice were bred and maintained in the AAALAC-accredited facility at Harvard Medical School at ambient temperature and humidity with 12 h light/dark cycles. Mice were specific pathogen-free (SPF) and maintained under a 12 h light/dark cycle with standard chow diet. Both male and female mice were used for all experiments, and mice were gender and age-matched within experiments. Experimental and control mice were co-housed whenever appropriate. All animal experiments were conducted in accordance with the guidelines of the Laboratory Animal Center of National Institutes of Health. The Institutional Animal Care and Use Committee of Harvard Medical School approved all animal protocols (IS111).

**Genotyping.** 564Igi mice were genotyped using digital droplet PCR (ddPCR). Tail DNA was isolated and digested with AluI (NEB). Droplets were prepared from a mix of tail DNA, primers (Supplementary Table 2), and EvaGreen Supermix (Bio-Rad) using a QX200 Droplet Generator (Bio-Rad). PCR was performed using a C100 Touch Thermal Cycler (Bio-Rad) and droplets were read on a QX200 Droplet Reader (Bio-Rad). 564Igi heavy (Hi) and light (Ki) chain copy number was quantified by comparing to amplification of reference mRPP30 using QuantaSoft (Bio-Rad). FACStyping of CD45.1 mice and bone marrow chimeras was performed by bleeding mice retroorbitally using heparinized capillary tubes and collecting into 30 μL of acid-citrate-dextrose solution (Sigma). Stabilized blood was underlayered with 1 mL of Lymphocyte Separation Medium (Corning) and centrifuged at $400 \times g$ for 30 min at room temperature. The mononuclear cell layer was aspirated and processed for flow cytometry as described below using anti-CD45.1 and anti-CD45.2 antibodies.

**Tissue collection**. Mice were sacrificed by cervical dislocation and blood was immediately collected via cardiac puncture. Mice were then transcardially perfused with 15–30 mL PBS and spleens and kidneys were dissected and allocated for immunofluorescence or flow cytometry. Blood was kept at room temperature for 1 h to promote coagulation, then centrifuged at $500 \times g$ for 10 min at 4 °C to isolate serum, which was stored at −80 °C.

**Bone marrow chimeras**. Epitope spreading was induced in an autoreactive B cell driven model of autoimmunity using mixed 564Igi chimeras as previously described[3]. Recipient mice between 8 and 10 weeks old were irradiated with 1100 rads and placed on water with sulfamethoxazole/trimethoprim for 10 days to prevent opportunistic infections. Femurs and tibia were dissected from 6 to 8-week-old congenic donor mice and rinsed through three rounds of HBSS supplemented with 10 mM HEPES, 1 mM EDTA, and 2% heat inactivated FBS. Bones were crushed in a mortar and passed through a 70 µm cell strainer (Corning). $5 \times 10^6$ 564Igi bone marrow cells and $10 \times 10^6$ WT bone marrow cells in 100 µL were injected intravenously into each irradiated recipient 8–10 h after irradiation. Six weeks after reconstitution chimerism was verified by FACStyping as described above.

**Immunization**. To generate germinal centers in non-autoimmune chimeras, mice were immunized intraperitoneally with 100 µg of 4-hydroxy-3-nitrophenylacetyl hapten conjugated to ovalbumin (NP-OVA, Biosearch) in 50 µL HBSS precipitated in 50 µL of Imject Alum (ThermoScientific) 6 weeks after irradiation and bone marrow reconstitution. Four weeks after immunization, mice received an intraperitoneal booster immunization of 100 µg of NP-OVA in 100 µL HBSS. For chronic immunization, mice were immunized with intraperitoneal NP-OVA in alum followed by booster immunization with NP-OVA every 3 weeks for up to 12 weeks total.

**Flow cytometry**. Spleens and lymph nodes were harvested into ice cold FACS buffer (PBS with 0.5% heat inactivated FBS and 0.05% sodium azide) and mechanically digested through a 70 µm cell strainer (Corning). Spleens were incubated in RBC lysis buffer (155 mM NH₄Cl, 12 mM NaHCO₃, 0.1 mM EDTA) for 3 min at room temperature and washed with FACS buffer. Cells were counted and $1 \times 10^6$ cells/well were added to round-bottom 96 well plates and incubated with 50 µL of staining mix (appropriate antibodies and viability dye in FACS buffer) for 30 min on ice. The following antibodies were used: anti-CD44 (IM7, 1:1000), anti-CD62L (MEL-14, 1:100), anti-CD45.1 (A20, 1:300), anti-CD45.2 (104, 1:300), anti-CXCR5 (L138D7, 1:200), anti-PD-1 (RMP1-30, 1:200), anti-Sca-1 (D7, 1:300), anti-GL7 (GL7, 1:200), anti-CD3 (500A2, 1:300), anti-CD4 (GK1.5, 1:300), anti-PSGL-1 (2PH1, 1:200), anti-GITR (DTA-1, 1:200), and anti-FoxP3 (FJK-16s, 1:100). Plates were washed with FACS buffer and for two-step staining procedures incubated with 50 µL of secondary staining mix (appropriate streptavidin antibody in FACS buffer) for 15 min on ice. For intracellular staining, cells were fixed with Fixation/Permeabilization Buffer (eBioscience) for 30 min at room temperature, washed with Permeabilization Buffer (eBioscience), and incubated with 50 µL of intracellular staining mix (appropriate intracellular antibody in Permeabilization Buffer) for 30 min at room temperature. Cells were washed with a final wash of FACS buffer, resuspended in 150 µL FACS buffer, and read using 3–8 fluorophore flow cytometry on a FACSCanto II (BD Biosciences) with 488, 405, and 640 nm lasers using FACSDiva (BD Biosciences). Compensation matrices were determined using unstained and single fluorophore stained controls. Data was analyzed using FlowJo (Tree Star).

**Cell sorting**. Cell suspensions were stained as for flow cytometry and resuspended in FACS buffer. For quantitative RT-PCR, a two-way semi-purity sort was performed using a SH-800Z (Sony) with 488, 405, 561, and 638 nm lasers. Cells were categorized as T_FH (CD45.2⁺CD45.1⁻CD3⁺CD4⁺CXCR-5⁺PD-1⁺GITR⁻) or T_FR (CD45.2⁺CD45.1⁻CD3⁺CD4⁺CXCR-5⁺PD-1⁺GITR⁺). Sorted cells were resuspended in TCL (Qiagen) and stored at −80 °C until RNA isolation. For droplet-based single cell sequencing, CD4⁺ cells were purified following flow cytometry staining using MACS CD4⁺ T Cell Isolation Kit (Miltenyi, 130-095-248) according to the manufacturer's protocols. A two-way purity sort was then performed using a FACSARIA II Special Order system (BD Biosciences) with 355, 405, 488, 640, and 592 nm lasers into PBS with 0.04% BSA. Cells were categorized as follicular T cells by CD45.2⁺CD45.1⁻CD3⁺CD4⁺CXCR-5⁺PD-1⁺.

**Droplet-based single-cell RNA and TCR sequencing**. The scRNA-seq and scTCR-seq libraries were prepared using the 10X Single Cell Immune Profiling Solution Kit (10X Genomics, #1000006, #1000020, #1000071, #1000152, #120262). Immediately post-sorting, cells were resuspended to final concentration of 100–800 cells per µL determined by hemocytometer. Cells were captured in droplets at a targeted recovery of 500–7000 cells and multiplet rate of 0.4–5.4% using a Chromium Controller (10X Genomics), followed by barcoding and reverse transcription. Emulsions were broken and cDNA was purified using Dynabeads MyOne SILANE and amplified by PCR. For gene expression library preparation, 2.4–50 ng of amplified cDNA was fragmented and end-repaired, double-sided size selected with SPRIselect beads (Beckman Coulter), PCR amplified with indexing primers,

and double-sided size-selected with SPRIselect beads. For TCR library construction, TCR transcripts were enriched from 2 µL of amplified cDNA by PCR, and 5–50 ng of PCR product was fragmented and end-repaired, size-selected with SPRIselect beads, PCR-amplified with indexing primers, and size-selected with SPRIselect beads. Sequencing of scRNA libraries were performed on a NextSeq 500 (Illumina) to a minimum sequencing depth of 15,000 reads per cell using read lengths of 26 bp read 1, 8 bp i7 index, 98 bp read 2 for the gene expression library, or a minimum sequencing of depth of 3000 reads per cell using read lengths of 150 bp read 1, 8 bp i7 index, 150 bp read 2 for the TCR library.

**Quantitative RT-PCR**. RNA was isolated by incubating cell lysates with 2 vol SPRIselect RNAClean XP beads (Beckman Coulter) for 10 min at room temperature. Samples were washed four times for 5 min with 80% ethanol in a magnetic field. Beads were air dried for 10 min at room temperature then resuspended in 16 µL DEPC H₂O for 10 min at room temperature. Beads were returned to the magnetic field and eluate was collected and quantified on a NanoDrop 1000 (Thermo Scientific). Samples were treated with iScript DNAse (Bio-Rad) and cDNA was synthesized using the iScript Reverse Transcription Supermix (Bio-Rad) according to manufacturer's protocols. Quantitative RT-PCR (qRT-PCR) was performed using Sso SYBR Green Supermix (Bio-Rad) and primers (Supplementary Table 3) on a CFX96 Touch Real-Time PCR Detection System (Bio-Rad) using QuantaSoft (Bio-Rad). All qRT-PCR reactions were performed in 10 µL and performed in duplicate or triplicate. Expression level was analyzed using comparative Ct and normalized to Ywhaz.

**Immunofluorescence confocal microscopy**. Isolated organs were individually perfused with PBS followed by 4% paraformaldehyde (PFA) in PBS. Tissues were fixed overnight in 4% PFA at 4 °C, cryoprotected with 30% sucrose in PBS for 8 h at 4 °C, perfused with 30% OCT (TissueTek), and embedded in 100% OCT in Standard Cryomolds (TissueTek) on dry ice and stored at −80 °C. Frozen sections were cut on a cryostat at a thickness of 16 µm and allowed to dry for 60 min at room temperature. Sections were fixed with 4% PFA in PBS for 10 min at room temperature, then permeabilized and blocked with 1% heat inactivated FBS in IF buffer (PBS with 1% BSA and 0.1% Triton X-100) for 1 h at room temperature. Slides were stained with primary antibody in IF buffer overnight at 4 °C, nuclei were counterstained with DAPI (5 µg/mL, Sigma) when indicated, and slides were mounted using Fluoro-Gel (Electron Microscopy Sciences). Images were acquired using a Fluoview FV100 inverted confocal microscope (Olympus) and analyzed using Fiji (ImageJ).

**Processing and filtering of scRNA-seq data**. The cellranger (10X Genomics, version 4.0.0) count pipeline was used to align 5' gene expression reads to the GRCm38 reference genome (mm10). Only barcodes with unique molecular identifier (UMI) counts that passed the threshold for cell detection were included in gene-barcode matrices. We obtained reads from 71,231 cells with an average of 1245 genes per cell and 26,367 reads per cell. Individual sample matrices were loaded in Seurat[80] (version 3.1.4) using the Read10X function and filtered for cells with at least 200 genes detected and genes detected in at least 3 cells using the CreateSeuratObject function, leaving 44,079 cells from WT chimeras and 27,139 cells from mixed 564Igi chimeras. Individual samples were merged using the merge function, and S and G2/M cell cycle phase scoring was assigned using CellCycleScoring. To remove batch effects between samples associated with a heat-shock gene expression signature, genes annotated with the Gene Ontology biological process (GOBP) term "cellular response to heat" (GO:0034605) was used to assign a heat shock score using AddModuleScore. Cells with less than 1000 or greater than 3500 genes detected, less than 2000 reads detected, greater than 7% mitochondrial RNA content, greater than 20% ribosomal RNA content, an S phase score greater than 0.15, or a G2/M phase score greater than 0.15 were excluded from analysis, with 15,280 cells from WT chimeras and 13,442 cells from mixed 564Igi chimeras passing the filters. BCR and TCR variable and constant genes were excluded from scRNA-seq analysis to prevent clustering based on VDJ transcripts. Genes *Gm42418* and *AY036118* were also removed, as they overlap the rRNA element Rn45s and represent rRNA contamination.

**Unsupervised clustering of scRNA-seq data**. Regularized negative binomial regression was performed on cells from WT or mixed 564Igi chimeras separately using the sctransform normalization method[81] to normalize, scale, select variable genes, and regress out mitochondrial RNA content, ribosomal RNA content, number of UMIs, and heat shock score. WT and mixed 564Igi chimera datasets were then integrated[82] using SelectIntegrationFeatures, PrepSCTIntegration, FindIntegrationAnchors, and IntegrateData. Following principal component analysis (PCA), clusters were identified using FindClusters to apply shared nearest neighbor (SNN)-based clustering using the first 25 principal components with $k = 30$ and resolution = 0.15. The same principal components were used to generate UMAP projections.

**Diffusion map and pseudotime analysis**. Seurat objects were exported to scanpy (version 1.5.1) using anndata2ri (version 1.0.2). Partition based graph abstraction was performed using the PAGA function[83] with 15 neighbors and the first 20 principal components. A randomly selected activated T_FH cell was used as the root cell for diffusion pseudotime computation using the first 10 diffusion components.

Diffusion component coordinates and pseudotime values were added back to the Seurat object using CreateDimReducObject and AddMetaData, respectively.

**Cell cluster annotation.** Clusters were annotated based on expression of marker genes for known populations and differentially expressed genes for novel populations, including *Foxp3* ($T_{FR}$), *S100a6* (activated $T_{FH}$), *Sostdc1* (Sostdc1 $T_{FH}$), *Sell, Slamf7* ($T_{FH}$-CM), *Ccl5, Gzmk* ($T_{FH}$-effector), *Ifit1, Isg15*, and *Ifi3* ($T_{FH}$-ISG). Cluster determination was confirmed by identifying differentially expressed marker genes for each cluster using FindAllMarkers with the MAST algorithm[84] and comparing to known cell-type specific marker genes. Cluster names were updated using RenameIdents.

**Differential gene expression.** Differentially expressed genes between autoimmune or non-autoimmune chimeras were determined using FindMarkers with the MAST algorithm across all genes. For differential expression analysis within individual clusters or clonotypes, Seurat objects were first subset using the subset function.

**Analysis of human scRNA-seq data.** Reference scRNA-seq data from human renal biopsies were obtained from ImmPort (SDY997)[10]. Raw count matrices were filtered for cells with at least 200 genes detected and genes detected in at least 3 cells using the CreateSeuratObject function, leaving 520 cells from healthy controls and 5627 cells from SLE patients. Cells with less than 1000 or greater than 3500 genes detected, or greater than 30% mitochondrial RNA content were excluded from analysis, with 162 cells from healthy controls and 2756 cells from SLE patients passing the filters. Regularized negative binomial regression was performed on cells from health controls or SLE patients separately using the sctransform normalization method to normalize, scale, select variable genes, and regress out mitochondrial RNA content and number of UMIs. Healthy controls and SLE datasets were then integrated using SelectIntegrationFeatures, PrepSCTIntegration, FindIntegrationAnchors, and IntegrateData. Following PCA, clusters were identified using FindClusters to apply SNN-based clustering using the first 25 principal components with $k = 30$ and resolution = 0.15. Clusters were annotated based on expression of marker genes for known populations, including *CD3E, CD4* (CD4 T cells), *CD8A* (CD8 T cells), *LYZ* (macrophages), *KLRF1* (NK cells), *MS4A1, BCL11A* (B cells), and *IRF8* (monocytes). CD4 and CD8 T cell clusters were subset using the subset function, leaving 93 cells from healthy controls and 1323 cells from SLE patients. Normalization and integration were performed as above using raw counts. Following PCA, clusters were identified using FindClusters to apply SNN-based clustering using the first 25 principal components with $k = 30$ and resolution = 0.7. Clusters were annotated based on expression of marker genes for known populations, including *CD4, CD40LG* ($T_{FH}$), *NKG7, GZMH, CD8A* (CD8), *TIGIT, IKZF2, FOXP3* (Treg), *IL2RB, ITGA1* (CD8-resident memory), *GZMK, EOMES, CCR5* (CD8-central memory), *ISG15, MX1*, and *OAS3* (CD4-ISG). Differentially expressed genes between healthy controls and SLE patients were determined using FindMarkers with the MAST algorithm across all genes. Comparison between human and mouse differential expression analysis was performed by determining the homologs of each gene using getLDS from biomaRt (version 2.24.0). From the 13,517 genes identified by scRNA-seq of mouse follicular T cells and 17,485 genes identified by scRNA-seq of human renal biopsies, 9576 homologs were identified in both datasets.

**Analysis of bulk human RNA-seq data.** Reference bulk RNA-seq data from human CD4$^+$ PBMCs were obtained from Bradley et al.[12]. Differentially expressed genes between healthy controls and SLE patients were determined using DEseq2 (version 1.26.0)[85]. Comparison between human and mouse differential expression analysis was performed by determining the homologs of each gene using getLDS from biomaRt (version 2.42.0)[86]. From the 13,517 genes identified by scRNA-seq of mouse follicular T cells and 17,752 genes identified by bulk RNA-seq data of human CD4$^+$ PBMCs, 10,962 homologs were identified in both datasets.

**Gene expression signature scoring.** Pathway analysis and gene set enrichment analysis was performed using clusterProfiler (version 3.14.3)[87]. Differentially expressed genes were selected using P-adj < 0.01 and ranked according to log$_2$FC for enrichment analysis. Ranked gene lists were used to query GOBP[88,89] and MSigDB (version 7.0.1)[90,91] signature libraries. Signature scores were assigned to individual cells, clonotypes, or specificity groups using AddModuleScore and gene lists from GOBP or MSigDB. For gene ontology analysis and annotation, differentially expressed genes were selected using P-adj < 0.05 and absolute log$_2$FC > 0.1 thresholds.

**Data processing of scTCR-seq libraries.** The cellranger (10X Genomics, version 4.0.0) vdj pipeline was used to align TCR reads to the vdj-GRCm38 alts ensemble 3.1.0 reference genome (10X Genomics). We obtained reads from 43,896 cells with an average of 9461 reads per cell. Only TCRs with full length and productive α and β chain sequences were included in analysis.

**TCR clonality analysis.** Clonotypes were determined by grouping cell barcodes that shared the same pair of productive CDR3α and CDR3β amino acid sequences,

and clone size was calculated by the number of unique cell barcodes belonging to an individual clonotype. Clonality was matched with gene-expression analysis in Seurat by adding clonality information to the metadata using AddMetaData based on cell barcodes. Unweighted TCR network analysis between samples and conditions was performed using the qgraph package (version 1.6.5). To evaluate public clonotype environments, non-metric multidimensional scaling (NMDS) was performed using vegan (version 2.5-6) with $k = 2$ for comparisons between autoimmune and non-autoimmune mice and with $k = 6$ for comparisons between clusters. Stress < 0.1 was confirmed using a Shepard plot. Clonotype Shannon diversity calculation and rarefaction analysis was performed using vegan with step size = 20. Public clonotypes were identified using immunarch (version 0.5.5).

**Unsupervised classification of TCR repertoires.** TCR repertoires of each individual chimera were constructed from CDR3αβ and VDJ gene usage from scTCR-seq. TCR featurization was performed using a variable autoencoder (VAE) in DeepTCR (version 1.4.15)[13] with 256 latent dimensions, $k = 5$ for the first convolutional layer of the graph, learned latent dimensionality scaling of 64 for amino acids, learned latent dimensionality of 48 for VDJ genes, latent alpha of 0.001, and three convolutional layers with 32, 64, and 128 neurons respectively. The VAE was trained using an Adam Optimizer with learning rate = 0.001 until convergence criteria of >0.01 decrease in determined interval was met. For sample-agnostic clustering, a dendrogram was constructed to compare clonotype distribution in UMAP space and PhenoGraph clustering using Kullback–Leibler divergence. TCR repertoire classification performance was assessed using a K-nearest neighbor (KNN) algorithm that was trained and tested following PhenoGraph clustering. K was varied from 1 to 16 in a 5-fold cross-validation strategy, where the predictive power was assessed using the left-out fold to calculate area under the curve (AUC).

**Clonal expansion and gene expression correlation analysis.** Spearman correlations between gene expression and continuous variables such as clonotype size and pseudotime were calculated across all cells or within individual clusters, as indicated. Genes were ranked according to correlation coefficients ($\rho$). To correlate average gene expression between individual clonotypes or specificity groups, the AverageExpression function from Seurat was used to calculate average raw counts of each individual gene across all cells belonging to each individual clonotype or specificity group.

**GLIPH2 analysis.** The grouping of lymphocyte interaction by paratope hotspots (GLIPH2) algorithm[14] was used to predict TCR specificity groups. GLIPH2 clusters TCRs based on a global similarity index, determined by CDR3 sequences that differ by up to one amino acid, and a local similarity index, determined by common CDR3 motifs of two to three amino acids. Global similarity was further restricted to TCR members of the same length and amino acid differences at the same position based on a BLOSUM62 matrix. Motifs with N or P encoded amino acids were given extra weight, and TCRs were allowed to be assigned to multiple clusters. The GLIPH2 mouse CD4 dataset was used for reference. Fisher's exact test was used to assess the statistical significance of a given motif, and specificity groups were filtered for clusters from at least four samples with significant V-gene bias ($P < 0.05$ by GLIPH2) and significant final score ($P < 1 \times 10^{-5}$ by GLIPH2). Specificity group prediction was matched with gene-expression analysis in Seurat using AddMetaData based on clonotype sequences.

**CDR3β database construction.** To predict antigen specificities, we created a single reference database of known CDR3β sequences and their cognate antigens by downloading published data from VDJdb[16], PIRD[17], and McPAS-TCR[18] (downloaded on April 8, 2020). Clone metadata including disease relevancy, protein name, peptide sequence, VDJ gene usage, associated publication, T cell type, and species information was cleaned to eliminate different conventions in annotation across datasets, and an additional metadata category of disease class (virus, bacteria, parasite, autoimmune, cancer, transplant, immunodeficiency, or allergy) was added. CDR3 sequences were limited to IUPAC letters and any duplicate entries were removed, resulting in a single reference database consisting of 99,809 unique CDR3β sequences with annotated specificities for 469 antigens and 482 peptides (Supplementary Data 4). To test whether TCR sequences identified by scTCR-seq had known antigen specifies, we searched for the presence of each CDR3β sequence in our reference database. To extend antigen predictions to CDR3β sequences not found in the reference database, we performed GLIPH2 analysis on our reference database, identifying annotated antigens associated with CDR3β local motif patterns. Antigen prediction was matched with gene-expression analysis in Seurat using AddMetaData based on GLIPH2 local motif predictions.

**Profile-based search for autoantigens.** To predict antigens using a peptide-directed approach, we identified all peptides associated with CDR3β sequences preferentially expanded in follicular T cells from autoimmune or non-autoimmune chimeras (clone size >3 and positive or negative log$_2$FC). Peptides that were 9 amino acids long were weighted by clone size x fold enrichment and used to create a positional-weighted matrix (PWM) using 2017PWM[92]. This matrix represents the enrichment ratio of each amino acid at a particular position in the peptide. Each PWM was used to score the mouse proteome (UniProt) using a 9-mer sliding search. P values and Bonferroni-corrected p values were calculated for each peptide, representing the probability of randomly selecting a peptide with fitness score as high as or higher than the scored peptide.

**Supervised classification of TCR sequences**. Peptides with at least 100 unique known CDR3β sequences from our reference database were used for supervised training using DeepTCR. TCR featurization was performed using CDR3β, V gene, and J gene usage embedding of a sequence classifier. Supervised training was performed using 5-fold Monte-Carlo cross-validation on the classifier with 5 kernels for the first convolutional layer of the graph, learned latent dimensionality of 64 for amino acids, learned latent dimensionality of 48 for VDJ genes, 12 nodes per fully-connected layer, and three convolutional layers with 32, 64, and 128 neurons respectively. Training was conducted on 75% of the data with learning rate = 0.001 and hinge-loss = 0 until convergence criteria of >0.001 decrease in determined interval was met. Peptide-based CDR3β classification model performance was assessed on the remaining 25% of data using a bootstrapping method to sample Monte-Carlo predictions with replacement 5000 times to approximate AUC. Features for all sequences were extracted from the latent space and used to perform hierarchical clustering of peptide annotations. This trained sequence classifier was used for sequence inference using CDR3β, V gene, and J gene usage of clonotypes identified by scTCR-seq. For repertoire-level visualization of inference, each clonotype was represented in a UMAP representation of the feature space of the training and validation datasets.

**Statistical analyses**. All values are expressed as mean ± SEM. We corrected for multiple comparisons and report adjusted *P* values using Bonferroni correction. For pathway analyses, Fisher's exact test was used with Benjamini–Hochberg correction for multiple testing. No statistical methods were used to predetermine sample size.

**Visualization**. Bar graphs were created using ggpubr (version 0.2.5.999) or Prism (GraphPad), venn diagrams were created using vennDiagram (version 1.6.20), and correlation vs correlation scatter plots were created using ggplot2 (version 3.3.0). Biological theme comparisons, network plots, and gene set enrichment plots were generated using clusterProfiler (version 3.14.3). Heatmaps and hierarchical clustering was performed using pheatmap (version 1.0.12). UMAP and violin plots comparing gene expression across samples and clusters were generated using Seurat. Sequence motifs were created using ggseqlogo (version 0.1). Scatter plots comparing gene expression and continuous variable were created using FeatureScatter from Seurat. Volcano plots were generated using EnhancedVolcano (version 1.4.0) and differentially expressed genes (absolute $\log_2 FC > 0.2$ and *P*-adj <0.01) were highlighted in red. Experimental diagrams and schematics were created with BioRender.com.

**Reporting summary**. Further information on research design is available in the Nature Research Reporting Summary linked to this article.

## Data availability

All scRNA-seq and scTCR-seq data generated in this study have been deposited in the GEO database and are available under primary accession number GSE157649. The following publicly available datasets were used: Immport SDY997, human SLE T cell RNA-seq[12] (https://doi.org/10.1371/journal.pone.0141171.s003), VDJdb, PIRD, McPAS-TCR, UniProt mouse proteome (https://www.uniprot.org/proteomes/UP000000589), and GRCm38 reference genome (https://www.ncbi.nlm.nih.gov/assembly/GCF_000001635.20/). Source data are provided with this paper.

## Code availability

Relevant code are available through github (https://github.com/egarren/scTfh).

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

## Acknowledgements

We thank H. Leung of the Optical Microscopy Core and J. Moore of the Flow and Imaging Cytometry Resource at the PCMM for technical assistance. E.A.G. was supported by NIH grants T32GM007753, T32AI007529, and F30AI160909. T.v.d.B. was supported by the H2020-MSCA-IF-GF project BEAT No. 796988. M.C.C. is supported by NIH R01AR074105.

## Author contributions

Conceptualization, E.A.G., T.v.d.B., L.S., and C.E.v.d.P.; Methodology, E.A.G., T.v.d.B., L.S., and C.E.v.d.P.; Software, E.A.G.; Validation, E.A.G.; Formal analysis, E.A.G.; Investigation, E.A.G., T.v.d.B., L.S., C.E.v.d.P., and C.C.; Data curation, E.A.G.; Writing, E.A.G.; Visualization, E.A.G.; Supervision, M.C.C.; Funding acquisition, M.C.C.

## Competing interests

The authors declare no competing interests.
