## [Peer Review File · Nature Communications]

Follicular T Cells are Clonally and Transcriptionally Distinct in B Cell-Driven Autoimmune DiseaseREVIEWER COMMENTS

Reviewer #1 (Remarks to the Author):

Akama-Garren et al. characterized and compared the transcriptome and TCR repertoire of follicular T cells induced by immunization and B cell-driven autoimmunity. Follicular T cells from autoimmune and non-autoimmune animals contained conserved subsets identified through unsupervised clustering with similar frequency distribution among these subsets. However, differentially expressed genes were identified between autoimmune and non-autoimmune conditions, suggestive of transcriptional reprogramming. Clonal analysis of follicular T cells revealed preferential public clonotype sharing among TCRs from autoimmune animals despite the lack of other distinguishing repertoire-wide features. Autoimmune conditions did not drive distinct patterns of expansion and clonality with the TCR repertoire compared to non-autoimmune, however, selected genes were differentially expressed in expanded clonotypes of the two conditions. Antigen binding predictions based on TCR sequences suggested a large overlap in antigen specificities between autoimmune and non-autoimmune animals including autoantigens, suggesting similar reactivities to foreign and self under the two conditions. The authors concluded that functional changes in follicular T cell subsets as suggested by their differential gene expression, and not inter-subset differences, may be causal to disease progression in this autoimmune model.

The idea to compare and contrast paired RNA and TCR sequencing between autoimmune and non-autoimmune responses is novel. The authors performed careful and extensive analyses despite the majority of their data shows a lack of significant differences between the two conditions. This observation is somewhat surprising given the authors' previous work, but may be valuable for future studies testing alternative theories. Their hypothesis that follicular T cells are permissive for tolerance loss and become dysfunctional upon encounter with autoreactive B cells is sensible, but by limiting their analyses to follicular T cells at one single timepoint, the authors have made the following assumptions without testing them: a) follicular T cells are responsible for driving autoimmunity, b) the responsible follicular T cells would still be present, abundant, and identifiable at their chosen timepoint, and c) any changes/dysregulation observed in follicular T cells at this time is the cause and not a downstream effect of autoimmunity. The findings presented in the manuscript suggest at least one of these assumptions to be false, the first two of which were identified by the authors and the third of which went unacknowledged. If the observed transcriptional changes are a result of the autoimmune landscape, for example the cytokine milieu, identifying factors that drive such remodeling as the authors suggested might not be conducive to uncovering the mechanism underlying autoimmune pathogenesis. Therefore, the authors should consider discussing the cause and effect relationship between autoimmune disease onset and follicular T cell dysfunction in their model.

Other points:

- The author cited previous work stating autoimmune B cells emerging from the germinal center reaction but a germinal center subset was not mentioned or identified in the follicular T cells using markers specific for same. The focus on the germinal center in this case is overstated.
- The data comparing non-autoimmune and lupus renal biopsies may not be comparable to the mouse data generated from lymphoid tissues. The tissue disparity issue aside, very few cells were collected from healthy controls, which might result in unreliable results from the differential gene analysis between healthy and autoimmune subsets.

Reviewer #2 (Remarks to the Author):

1. Row 107f. ad

"We performed pseudotime analysis to determine potential developmental trajectories between follicular T cell subsets, revealing that central memory, effector, and ISG subsets represent distinct states in pseudotime and on a diffusion map (Fig. 1d)."

The authors should explain in more detail what they plot in fig. 1d on the right side. Is this just pseudo-time projected in other dimensions?

2. Suppl. Fig. 2d and also mentioned in the line 141f: Is this representative with such a difference in sample size of controls and diseased? Absolute numbers are not mentioned, so this should be clarified.

3. Fig. 2e: authors will need to specify what is the cutoff for these genes used for the gene ontology analysis?

4. Fig. 3a: authors should explicitly mention where edges are drawn in this network plot

5. row 206: NMDS analysis seems creative to use this method, but I am not very familiar with it. The authors should thus add more detail, including mentioning what exactly is plotted in the Fig. 3e or how it could be interpreted.

6. Fig 3g: Would be important to know if within the expanded clonotypes, how is the cell type distribution? e.g. in clone with 150 cells, what percentage are TFR etc?

7. Fig. 4b: why are there 1.5 samples? Are these duplicates in the mouse experiments?  this should be mentioned in the caption. The authors also never define what public clone/clonotype means, as there are varying definitions in the field this needs to be specified. There are also black and grey circles, which is also not clarified in the caption.

8. Fig 5d: I think there should be consistency in how the authors make the logo plots (I think the frequency one is better; for some of them it's really difficult to read the letters). I also think numbering the residues would help for easier comparison; because the 1st one is longer, but has a space between 2 residues.

9. Suppl. Fig 5e: still seems as if many specificity groups are unique for Tfh-act & Tfr? Maybe this is just resolution issue, but seems a bit misleading, since the Venn overlap plot says otherwise.

10. Row 311f: "CDR3 motif analysis revealed that both shared and condition-specific specificity groups were largely determined by contact residues in the CDR3 β sequence (Fig. 5d), which is the basis for antigen-specific paratope convergence" Which are these residues and this seems like an overgeneralization. The authors should qualify this for at least peptides that GLIPH has demonstrated specificity based on experimental validation.

11. Row 318: "All large public specificity groups were polyclonal with limited clonotype sharing amongst" which groups do they refer to? What's the definition of 'large public'?

12. A major concern is the text from the final results section and corresponding to Figure 7 and the claims on predicting T cell reactivities is far too unsubstantiated. While it is okay to suggest that there are groupings of similar CDR3B sequences that suggest similar specificity, without experimental validation (cloning, expressing and testing of TCRs for binding to target antigen, as was done by Glanville et al., Nature 2017, Figure 5e), there is no way to be convinced that these groupings truly reflect antigen specificity.

The concern is also related to the fact that by basing this analysis only on CDR3B, the authors assume that a similar CDR3B guarantees similar specificity. While this might be the case for some TCR-peptide-MHC pairings, I cannot accept that it would be true all the time.

Overall, this section just has too many overstatements and assumptions on TCR specificity without proper validation, these types of generalizations are not good for the field and I suggest the authors substantially modify and remove most of these strong claims from the section.

Minor comments

Fig 1b: Legend to the colors on the heatmap is missing. Perhaps the legend from fig.1f should be applied to other plots, should be mentioned in the figure caption

Suppl.Fig 3g: figure is not readable at this resolution, should be addressed in final version.

Row 333: they refer to Fig. 5g, but there is not Fig. 5g; Maybe they mean Suppl. Fig. 5i?

Reviewer #3 (Remarks to the Author):

The manuscript "Follicular T cells are clonally and transcriptionally distinct in b cell-driven autoimmune disease" investigated diversity and functional difference of Tfh cells in autoreactive B cell-driven animal models by using single-cell RNA-seq and single cell TCR repertoire analysis. Although key functions of Tfh in autoimmune disease are universally accepted, precise mechanism whether autoreactive Tfh cells are functionally distinct in disease condition or whether autoreactive Tfh cells exist in disease condition is not clearly understood. To answer these questions, current study investigated clonality and function of Tfh cells by single-cell level of RNA seq and paired ab TCR repertoire analysis in established animal model. Author made several interesting observations: 1) Tfh cells are transcriptionally distinct in autoimmune disease, 2) clonotypic expansion of Tfh between autoimmune and non-autoimmune is not different but clonal expansion is associated with differential gene expression, 3) antigen specificity between autoimmune and non-autoimmune Tfh cells are shared, and 4) minor group of autoreactive Tfh groups are transcriptionally distinct.

Observations are interesting and most studies and data presented in the manuscript are convincing. Some concerns should be addressed before publication.

1. One of major limitation is the experimental setting to generate Tfh cells from autoimmune and non-autoimmune potentially generate biased results. Differentiation time (12 weeks of autoimmune and 2-4 weeks immunization-induced non-autoimmune Tfh) is not controlled properly. It is not shown whether the reconstitution of immune cells and immune profile in two groups of reconstituted mice. Author did not explain why they need boost immunization to harvest Tfh cells? Some observations (for example, increased Tfr proportion in autoimmune mice compared to non-autoimmune mice) potentially driven by experimental setting not precisely due to the biology.

2. Authors analyzed clonality and transcriptome in a single cell level thoroughly. However, no biological meaning was not discussed. It is not clear whether shared transcriptome, specificity in specific cluster of Tfh in autoimmune vs. non-autoimmune (figure 5) suggest pathogenic Tfh?

3. It is surprisingly there is no expansion of anti-ribosomal antigen-reactive Tfh in autoimmune mice.

4. It is not clear the sentence in discussion (page 24 lines 491-493).

5. Only increased Sca-1 expression was confirmed. Conclusions should be confirmed in different cohort.

REVIEWER COMMENTS

We would like to thank all three reviewers for their thoughtful summary and comments. The reviewers raise important critiques that we have addressed comprehensively below. In addition to depositing our data in publicly available repositories (GSE157649), we have also created a GitHub repository to host the relevant code used for this manuscript (<https://github.com/Carroll-Michael/scTfh>) and have completed and attached the relevant submission checklists.

Reviewer #1 (Remarks to the Author):

Akama-Garren et al. characterized and compared the transcriptome and TCR repertoire of follicular T cells induced by immunization and B cell-driven autoimmunity. Follicular T cells from autoimmune and non-autoimmune animals contained conserved subsets identified through unsupervised clustering with similar frequency distribution among these subsets. However, differentially expressed genes were identified between autoimmune and non-autoimmune conditions, suggestive of transcriptional reprogramming. Clonal analysis of follicular T cells revealed preferential public clonotype sharing among TCRs from autoimmune animals despite the lack of other distinguishing repertoire-wide features. Autoimmune conditions did not drive distinct patterns of expansion and clonality with the TCR repertoire compared to non-autoimmune, however, selected genes were differentially expressed in expanded clonotypes of the two conditions. Antigen binding predictions based on TCR sequences suggested a large overlap in antigen specificities between autoimmune and non-autoimmune animals including autoantigens, suggesting similar reactivities to foreign and self under the two conditions. The authors concluded that functional changes in follicular T cell subsets as suggested by their differential gene expression, and not inter-subset differences, may be causal to disease progression in this autoimmune model.

The idea to compare and contrast paired RNA and TCR sequencing between autoimmune and non-autoimmune responses is novel. The authors performed careful and extensive analyses despite the majority of their data shows a lack of significant differences between the two conditions. This observation is somewhat surprising given the authors' previous work, but may be valuable for future studies testing alternative theories. Their hypothesis that follicular T cells are permissive for tolerance loss and become dysfunctional upon encounter with autoreactive B cells is sensible, but by limiting their analyses to follicular T cells at one single timepoint, the authors have made the following assumptions without testing them: a) follicular T cells are responsible for driving autoimmunity, b) the responsible follicular T cells would still be present, abundant, and identifiable at their chosen timepoint, and c) any changes/dysregulation observed in follicular T cells at this time is the cause and not a downstream effect of autoimmunity. The findings presented in the manuscript suggest at least one of these assumptions to be false, the first two of which were identified by the authors and the third of which went unacknowledged. If the observed transcriptional changes are a result of the autoimmune landscape, for example the

cytokine milieu, identifying factors that drive such remodeling as the authors suggested might not be conducive to uncovering the mechanism underlying autoimmune pathogenesis. Therefore, the authors should consider discussing the cause and effect relationship between autoimmune disease onset and follicular T cell dysfunction in their model.

We agree that by only profiling follicular T cells at a single time point we are limited in the conclusions we can make and agree with the reviewer's list of assumptions that have been made. We also agree that the alterations in follicular T cell expression that we have observed might be a consequence, rather than the cause, of the autoimmune environment. Indeed, we believe this is likely the case for one of the differentially expressed genes we identified, *Ly6a*. In the text we acknowledged this by referencing studies that identified *Ly6a* as an interferon response gene in the discussion. In this revision we have added to this by explicitly stating this might be driven by the autoreactive environment and cytokine milieu, rather than reflecting cell-autonomous dysfunction. We have also added statements to the discussion explicitly acknowledging the reviewer's third assumption.

Other points:

- The author cited previous work stating autoimmune B cells emerging from the germinal center reaction but a germinal center subset was not mentioned or identified in the follicular T cells using markers specific for same. The focus on the germinal center in this case is overstated.

Single cell sequencing was performed on follicular T cells sorted from the spleens of mixed bone marrow chimera mice. We used surface expression of CXCR-5 and PD-1 to sort follicular T cells (Fig. 1a and 2c). CXCR-5 and PD-1 are widely accepted markers of follicular T cells (<https://pubmed.ncbi.nlm.nih.gov/17911595/>, <https://www.ncbi.nlm.nih.gov/pmc/articles/PMC3280079/>). Therefore, all follicular T cells in this study are assumed to have originated from germinal center reactions. The subsets annotated by scRNA-seq here represent subpopulations within this germinal center population.

- The data comparing non-autoimmune and lupus renal biopsies may not be comparable to the mouse data generated from lymphoid tissues. The tissue disparity issue aside, very few cells were collected from healthy controls, which might result in unreliable results from the differential gene analysis between healthy and autoimmune subsets.

We agree that comparisons of our bone marrow chimera data and the human data is limited by different species, different tissues, and the sparsity of T cells in healthy control kidneys. However, the purpose of this experiment was to simply validate whether some of the gene expression changes observed in mice are

recapitulated in human. As we show, even with these limitations, we observe a set of genes that are commonly differentially expressed in multiple datasets. We have added a note to our results section acknowledging the difficulty of performing differential expression with the limited T cells from healthy controls.

Reviewer #2 (Remarks to the Author):

1. Row 107f. ad

“We performed pseudotime analysis to determine potential developmental trajectories between follicular T cell subsets, revealing that central memory, effector, and ISG subsets represent distinct states in pseudotime and on a diffusion map (Fig. 1d).”

The authors should explain in more detail what they plot in fig. 1d on the right side. Is this just pseudo-time projected in other dimensions?

The right plot in Fig. 1d is a diffusion map, which plots each cell in two-dimensional space according to two diffusion components. The cells are colored by their cluster annotation from Fig. 1b. Pseudotime was calculated using a graph adjacency matrix and pseudotime values were plotted on the original UMAP in the left plot of Fig. 1d. Practically, the right plot of Fig. 1d can be interpreted as a graphical representation of the branching trajectory. This method is described in Wolf et al. *Genome Biology* (2018). We have clarified this by adding detail to the results and figure legend.

2. Suppl. Fig. 2d and also mentioned in the line 141f: Is this representative with such a difference in sample size of controls and diseased? Absolute numbers are not mentioned, so this should be clarified.

Yes, this is correct, unsurprisingly there are fewer infiltrating T cells in the kidneys of healthy controls than SLE patients. We have added an acknowledgement of this limitation in the results text. The purpose of this experiment was to compare gene expression changes in mice and human and despite this limitation, we observe a set of genes that are commonly differentially expressed in multiple datasets. See also response to Reviewer #1.

3. Fig. 2e: authors will need to specify what is the cutoff for these genes used for the gene ontology analysis?

A logFC threshold of 0.1 and adjusted p-value threshold of 0.05 was used for genes included in this analysis. This has been clarified in the methods.

4. Fig. 3a: authors should explicitly mention where edges are drawn in this network plot

Edges indicate clonotype membership to individual mice. In other words, that a given clonotype was identified in the connected mouse. This has been clarified in the figure legend.

5. row 206: NMDS analysis seems creative to use this method, but I am not very familiar with it. The authors should thus add more detail, including mentioning what exactly is plotted in the Fig. 3e or how it could be interpreted.

Non-metric multidimensional scaling is typically used in ecology research to determine how similar or different communities are based on species sharing between the communities (https://link.springer.com/chapter/10.1007/978-94-009-4061-1_9, <https://cran.r-project.org/web/packages/vegan/vegan.pdf>). We adapted this approach to immunologic clonotypic comparisons by treating each mouse as a different community and each clonotype as a different species. Clonotypes only present in an individual mouse will only be attributed to that mouse, whereas clonotypes present in multiple mice (public clonotypes) will be graphed in NMDS space based on their sharing amongst the different mice. These public clonotypes are used to determine the locations of individual mice in NMDS space, which can then be used graphically to determine sample similarity. Similar implementations of NMDS have been used in prior deep sequencing analyses, for example when comparing microbiota diversity (<https://www.nature.com/articles/s41598-017-06665-3>). We have added to the results text to clarify this method.

6. Fig 3g: Would be important to know if within the expanded clonotypes, how is the cell type distribution? e.g. in clone with 150 cells, what percentage are TFR etc?

The cluster distribution of individual expanded clonotypes is shown in Supplementary Fig. 3i. Each column represents an individual clonotype and each row represents an individual cluster. The color of each heatmap cell represents which percent of cells for that clonotype belong to that cluster. Clone size is also shown in the second row.

7. Fig. 4b: why are there 1.5 samples? Are these duplicates in the mouse experiments?  this should be mentioned in the caption. The authors also never define what public clone/clonotype means, as there are varying definitions in the field this needs to be specified. There are also black and grey circles, which is also not clarified in the caption.

None of the points in Fig. 4b represent 1.5 samples (that was just shown in the legend due to our plotting software). We have updated the legend to only reflect the possible sizes shown in the plot. Throughout our manuscript, public clonotypes refer to clonotypes present in more than one chimera. This is defined on pg. 11 in the sentence, "...given the presence of sparse clonotypes shared amongst individual chimeras, we next examined whether these public clonotypes might be capable of distinguishing autoimmune and non-autoimmune follicular T cells." In Fig. 4b the darker circles are simply points where multiple grey circles overlap with each other.

8. Fig 5d: I think there should be consistency in how the authors make the logo plots (I think the frequency one is better; for some of them it's really difficult to read the letters). I also think numbering the residues would help for easier comparison; because the 1st one is longer, but has a space between 2 residues.

Thank you for these suggestions. We have recreated Fig. 5d by making logo plots representing frequency and including numbering for each residue. We also updated the logo plots in Suppl. Fig 3 and Suppl. Fig 4 according to these guidelines.

9. Suppl. Fig 5e: still seems as if many specificity groups are unique for Tfh-act & Tfr? Maybe this is just resolution issue, but seems a bit misleading, since the Venn overlap plot says otherwise.

In Suppl Fig 5e each of the smaller circles represent a single clonotype but note than an individual clonotype may belong to multiple clusters. Indeed, we know this is the case from Suppl Fig 3i. This is not possible to represent in the network plot and we have instead colored each clonotype according to its most prevalent cluster annotation derived from scRNA-seq data. We have clarified this in the figure legend.

10. Row 311f: “CDR3 motif analysis revealed that both shared and condition-specific specificity groups were largely determined by contact residues in the CDR3 β sequence (Fig. 5d), which is the basis for antigen-specific paratope convergence” Which are these residues and this seems like an overgeneralization. The authors should qualify this for at least peptides that GLIPH has demonstrated specificity based on experimental validation.

GLIPH uses contact motif prediction for the basis of its algorithm, but we agree that without experimental validation we cannot know with certainty that the motifs identified by GLIPH from our scTCR-seq data are indeed contact residues. We have therefore modified the referenced sentence to read, “CDR3 motif analysis identified unique motifs in both shared and condition-specific specificity groups (Fig. 5d) that are predicted to be contact residues by antigen-specific paratope convergence.”

11. Row 318: “All large public specificity groups were polyclonal with limited clonotype sharing amongst” which groups do they refer to? What’s the definition of ‘large public’?

Large public specificity groups were defined using an absolute $\log_2FC < 2.5$ and total size > 10 . However, we corrected the above sentence to read, “Public specificity groups (absolute $\log_2FC < 2.5$) were often polyclonal ...” which is best appreciated in Fig 5b, which we have also added a reference to in the text.

12. A major concern is the text from the final results section and corresponding to Figure 7 and the claims on predicting T cell reactivities is far too unsubstantiated. While it is okay to suggest that there are groupings of similar CDR3B sequences that suggest similar specificity, without experimental validation (cloning, expressing and testing of TCRs for binding to target antigen, as was done by Glanville et al., Nature 2017, Figure 5e), there is no way to be convinced that these groupings truly reflect antigen specificity.

The concern is also related to the fact that by basing this analysis only on CDR3B, the authors assume that a similar CDR3B guarantees similar specificity. While this might be the case for some TCR-peptide-MHC pairings, I cannot accept that it would be true all the time.

Overall, this section just has too many overstatements and assumptions on TCR specificity without proper validation, these types of generalizations are not good for the field and I suggest the authors substantially modify and remove most of these strong claims from the section.

We agree that a major limitation of our study, and of this section in particular, is lack of validation of predicted antigenic targets. While these are questions we are certainly interested in, we believe the main scope of this study is on the technological advances and biological conclusions we are able to make from the scRNA-seq and scTCR-seq data. We agree that the antigenic predictions made are speculative and have addressed this in the text by modifying our conclusions to avoid overstatements. We have removed strong claims of antigen specificity from this section as well. While we agree that CDR3b sequence alone is possibly insufficient to predict TCR specificity, we are limited to using the CDR3b sequence alone as the public databases of TCR specificity we used to match TCR sequences and to train our machine learning model only have CDR3b sequence data.

Minor comments

Fig 1b: Legend to the colors on the heatmap is missing. Perhaps the legend from fig.1f should be applied to other plots, should be mentioned in the figure caption

We have added legends for the clusters to Fig.1b, 1e, and 1f.

Suppl.Fig 3g: figure is not readable at this resolution, should be addressed in final version.

We have enlarged the text in the venn diagram to make it more readable.

Row 333: they refer to Fig. 5g, but there is not Fig. 5g; Maybe they mean Suppl. Fig. 5i?

Yes, thank you, we have corrected this reference in the text.

Reviewer #3 (Remarks to the Author):

The manuscript "Follicular T cells are clonally and transcriptionally distinct in B cell-driven autoimmune disease" investigated diversity and functional difference of Tfh cells in autoreactive B cell-driven animal models by using single-cell RNA-seq and single cell TCR repertoire analysis. Although key functions of Tfh in autoimmune disease are universally accepted, precise mechanism whether autoreactive Tfh cells are functionally distinct in disease condition or whether autoreactive Tfh cells exist in disease condition is not clearly understood. To answer these questions, current study investigated clonality and function of Tfh cells by single-cell level of RNA seq and paired ab TCR repertoire analysis in established animal model. Author made several interesting observations: 1) Tfh cells are transcriptionally distinct in autoimmune disease, 2) clonotypic expansion of Tfh between autoimmune and non-autoimmune is not different but clonal expansion is associated with differential gene expression, 3) antigen specificity between autoimmune and non-autoimmune Tfh cells are shared, and 4) minor group of autoreactive Tfh groups are transcriptionally distinct.

Observations are interesting and most studies and data presented in the manuscript are convincing. Some concerns should be addressed before publication.

1. One of major limitation is the experimental setting to generate Tfh cells from autoimmune and non-autoimmune potentially generate biased results. Differentiation time (12 weeks of autoimmune and 2-4 weeks immunization-induced non-autoimmune Tfh) is not controlled properly. It is not shown whether the reconstitution of immune cells and immune profile in two groups of reconstituted mice. Author did not explain why they need boost immunization to harvest Tfh cells? Some observations (for example, increased Tfr proportion in autoimmune mice compared to non-autoimmune mice) potentially driven by experimental setting not precisely due to the biology.

We agree that the kinetics of the autoimmune and immunization conditions are different, which may account for the differences observed. As a point of clarification, the non-autoimmune controls received a total of 6 weeks of immunization, not 2-4 weeks. We waited for 6 weeks after irradiation before beginning immunization in order to give both autoimmune and non-autoimmune mice sufficient time to reconstitute their bone marrow. Both autoimmune and non-autoimmune mice were sacrificed 12 weeks after irradiation. Additionally, "increased Tfr proportion in autoimmune mice" was not something we observed in our model.

We believe that even if the observed differences in Tfh expression are due to differences in immune response kinetics, they still reflect fundamental differences between an autoimmune response and response to immunization, as these responses necessarily occur with different chronicity both in mouse models and in human disease. We chose to harvest follicular T cells two weeks after immunization boost because beyond that time period the immune response will resolve, which does not occur in autoimmune disease. We have extensively characterized the reconstitution and immune profiles of our mixed bone marrow

chimera model in Degn et al. *Cell* (2017). We have added a qualifying statement to the discussion acknowledging the potential role of immune response kinetics to our observed findings.

Furthermore, to address this reviewer's concern, we have performed two additional time course experiments to directly compare Tfh and Tfr cells in autoimmune and non-autoimmune mice at matched time points. Wild type or autoimmune mice were analyzed at 6 or 12 weeks and expression of Sca-1 on follicular T cells was analyzed by flow cytometry. In the first experiment, we used the mesenteric lymph nodes as a source of chronic foreign antigen. Mesenteric lymph nodes are sites of tolerance induction to food proteins and help prevent commensal bacteria from systemic exposure, effectively serving as a secondary lymphoid organ site of continuous exposure to foreign antigens (<https://www.ncbi.nlm.nih.gov/pmc/articles/PMC2118258/>). In the second time course experiment, wild type mice were continuously immunized for up to 12 weeks with foreign antigen as a source of chronic antigen. After initial intraperitoneal immunization with NP-OVA in alum, mice were boosted with NP-OVA in HBSS every 3 weeks. In both experiments, flow cytometry analysis revealed that CD4, Tfh, and Tfr cells from autoimmune mice expressed elevated Sca-1 regardless of length or route of exposure to antigen (Supplemental Fig 2).

We have added a sentence to our Results section describing these experiments. Based on these findings, in addition to our previous characterization of immune profiles in reconstituted mice (Degn et al. *Cell* 2017), we believe that our observations reflect differences due to an autoimmune environment, and not differentiation time.

2. Authors analyzed clonality and transcriptome in a single cell level thoroughly. However, no biological meaning was not discussed. It is not clear whether shared transcriptome, specificity in specific cluster of Tfh in autoimmune vs. non-autoimmune (figure 5) suggest pathogenic Tfh?

In the discussion we hypothesize that the shared predicted specificities between autoimmune and non-autoimmune conditions might represent bystander activation of follicular T cells. Ritvo et al. *PNAS* (2018) found that immunization did not significantly alter Tfh or Tfr clonality but rather resulted in non-specific Tfh bystander activation possibly due to TCR cross-reactivity. We propose that overlap in predicted specificity in our model represents bystander activation due to the highly cross-reactive nature of the TCR, particularly given the high diversity and small clone size observed in public specificity groups (Fig. 5f). Biologically, this leads to the hypothesis that just as dysfunctional Tfh cells might help autoreactive B cells overcome clonal anergy, autoreactive B cells might help autoreactive Tfh cells overcome T cell ignorance.

3. It is surprisingly there is no expansion of anti-ribosomal antigen-reactive Tfh in autoimmune mice.

Our antigen prediction experiments were done by creating a reference database of CDR3b sequences with known specificities from published studies. This database included 469 unique antigens, but none of these antigens were ribosomal antigens. In other words, within the TCR databases we queried, there were no published sequences of TCRs that are bind ribosomal antigens, and we are therefore unable to predict this specificity using this approach. Indeed, few of the autoimmune enriched specificity groups were successfully annotated, suggesting that the antigens to which autoimmune follicular T cells preferentially respond are not present in publicly available databases.

4. It is not clear the sentence in discussion (page 24 lines 491-493).

We have rewritten this sentence to read, “Functional differences within Tfr, Tfh, and Sostdc1+ Tfh cells, rather than differences in the relative frequencies of these subsets, are therefore likely responsible for the autoimmunity observed in our mixed chimera model of autoantibody-mediated disease.”

5. Only increased Sca-1 expression was confirmed. Conclusions should be confirmed in different cohort.

We validated gene expression changes in a separate cohort by sorting follicular T cells and performing qRT-PCR for a subset of differentially expressed genes in Suppl Fig 2a and Suppl Fig 2b. For example, we also validated increased expression of the lncRNA *Gm42031* in this experiment.

REVIEWERS' COMMENTS

Reviewer #1 (Remarks to the Author):

The original summary of this paper stands. The reviewers have addressed my concerns save one.

It is important to note in the text that the Tfh cells examined here have not necessarily "originated from germinal center reactions", by contrast to the authors' rather definitive rebuttal that they have. In fact, it is quite likely some have not originated from the GC, a point buttressed the authors' data. For example, the memory Tfh cells identified have likely not passed through the GC (He and Yu, *Immunity*, 2013). CXCR5 and PD-1 expression arise on cells destined, or not, for the GC, prior to GC entry (Kerfoot and Haberman, *Immunity*, 2011). GC Tfh cells are distinguishable from pre-follicular and follicular Tfh cells by differential expression of PD-1 and CXCR5 (Yusuf and Crotty, *JCI*, 2010), which the flow plot (2C) does not attempt to discern. Extrafollicular Tfh cells that provide B help are well described, and lack any evidence they are GC derived (Lee and Vinuesa, *JEM*, 2011). And so on, as evidenced in several well written reviews of the topic. This is not a trivial point, as it gets to the heart of the original concern raised about limiting their analysis to a single timepoints and kinetics.

Reviewer #2 (Remarks to the Author):

The authors have improved the clarity and quality of their manuscript with the revised version.

Reviewer #3 (Remarks to the Author):

This is the revised manuscript followed by reviewer's critiques. Authors responded to the points raised from the initial review. Major critique was the disparity of time points between autoimmune and non-autoimmune mice. Although it was not perfectly addressed, authors increased immune activation and subsequent Tfh cell activation in non-autoimmune mice. Inclusion of Tfh cells from MLNs is useful, too. Author also addressed other minor points. All the new data and discussions are included in the revised manuscript.

REVIEWERS' COMMENTS

We would like to thank the reviewers and editors for reviewing our revised manuscript. We are encouraged by their positive comments and have revised our manuscript to address their remaining concerns, in addition to addressing them point by point below.

Reviewer #1 (Remarks to the Author):

The original summary of this paper stands. The reviewers have addressed my concerns save one.

It is important to note in the text that the Tfh cells examined here have not necessarily "originated from germinal center reactions", by contrast to the authors' rather definitive rebuttal that they have. In fact, it is quite likely some have not originated from the GC, a point buttressed the authors' data. For example, the memory Tfh cells identified have likely not passed through the GC (He and Yu, Immunity, 2013). CXCR5 and PD-1 expression arise on cells destined, or not, for the GC, prior to GC entry (Kerfoot and Haberman, Immunity, 2011). GC Tfh cells are distinguishable from pre-follicular and follicular Tfh cells by differential expression of PD-1 and CXCR5 (Yusuf and Crotty, JI, 2010), which the flow plot (2C) does not attempt to discern. Extrafollicular Tfh cells that provide B help are well described, and lack any evidence they are GC derived (Lee and Vinuesa, JEM, 2011). And so on, as evidenced in several well written reviews of the topic. This is not a trivial point, as it gets to the heart of the original concern raised about limiting their analysis to a single timepoints and kinetics.

Thank you for this important clarification of Tfh fate and GC entry. We agree that CXCR-5 and PD-1 expression cannot absolutely distinguish GC origin of Tfh cells. Considering this significant distinction, we do not claim in the text that these cells have originated from GC reactions. However, to clarify this point further, we have added a statement to the discussion describing the potential non-GC origins of these cells.

Reviewer #2 (Remarks to the Author):

The authors have improved the clarity and quality of their manuscript with the revised version.

Thank you.

Reviewer #3 (Remarks to the Author):

This is the revised manuscript followed by reviewer's critiques. Authors responded to the points raised from the initial review. Major critique was the disparity of time points between autoimmune and non-autoimmune mice. Although it was not perfectly addressed, authors increased immune activation and subsequent Tfh cell activation in non-autoimmune mice. Inclusion of Tfh cells from MLNs is useful, too. Author also addressed other minor points. All the new data and discussions are included in the revised manuscript.

Thank you.